# UNC93B1 mediates differential trafficking of endosomal TLRs

Bettina L Lee[1], Joanne E Moon[1], Jeffrey H Shu[1], Lin Yuan[2], Zachary R Newman[1], Randy Schekman[2,3], Gregory M Barton[1,2]*

[1]Division of Immunology and Pathogenesis, Department of Molecular and Cell Biology, University of California, Berkeley, Berkeley, United States; [2]Division of Cell and Developmental Biology, Department of Molecular and Cell Biology, University of California, Berkeley, Berkeley, United States; [3]Howard Hughes Medical Institute, University of California, Berkeley, Berkeley, United States

**Abstract** UNC93B1, a multipass transmembrane protein required for TLR3, TLR7, TLR9, TLR11, TLR12, and TLR13 function, controls trafficking of TLRs from the endoplasmic reticulum (ER) to endolysosomes. The mechanisms by which UNC93B1 mediates these regulatory effects remain unclear. Here, we demonstrate that UNC93B1 enters the secretory pathway and directly controls the packaging of TLRs into COPII vesicles that bud from the ER. Unlike other COPII loading factors, UNC93B1 remains associated with the TLRs through post-Golgi sorting steps. Unexpectedly, these steps are different among endosomal TLRs. TLR9 requires UNC93B1-mediated recruitment of adaptor protein complex 2 (AP-2) for delivery to endolysosomes while TLR7, TLR11, TLR12, and TLR13 utilize alternative trafficking pathways. Thus, our study describes a mechanism for differential sorting of endosomal TLRs by UNC93B1, which may explain the distinct roles played by these receptors in certain autoimmune diseases.

## Introduction

Toll-like receptors (TLRs) recognize conserved microbial features and initiate signals critical for induction of immune responses to infection. A subset of TLRs (TLR3, TLR7, TLR8, and TLR9) recognizes forms of nucleic acids, including double-stranded RNA, single-stranded RNA, and DNA (*Barbalat et al., 2011*). This specificity facilitates recognition of a broad array of microbes but introduces the potential for recognition of self-nucleic acids. TLR7 and TLR9 recognition of self-RNA and self-DNA, respectively, contributes to autoimmune diseases such as systemic lupus erythematosus (SLE) (*Marshak-Rothstein, 2006*; *Christensen and Shlomchik, 2007*).

Discrimination between self and microbial nucleic acids cannot be achieved solely through recognition of distinct features but instead relies on differential delivery of these potential ligands to TLRs (*Barton and Kagan, 2009*). All of the TLRs capable of nucleic acid recognition localize within endosomal compartments which sequesters these receptors away from self nucleic acids in the extracellular space (*Barton and Kagan, 2009*). Our previous studies as well as work from other groups indicate that a requirement for ectodomain cleavage of intracellular TLRs further restricts receptor activation to protease-rich acidic compartments (*Ewald et al., 2008, 2011*; *Park et al., 2008*; *Garcia-Cattaneo et al., 2012*). Bypassing this requirement enables responses to extracellular self nucleic acid and leads to fatal inflammatory disease in mice (*Mouchess et al., 2011*). Moreover, the system appears carefully balanced as simply overexpressing TLR7 in mice causes responses to self-RNA and development of an SLE-like disease (*Pisitkun et al., 2006*; *Subramanian et al., 2006*; *Deane et al., 2007*). Thus, defining the regulatory steps that control TLR localization and influence the threshold of receptor activation has important implications for self/non-self discrimination.

*For correspondence: barton@berkeley.edu

**eLife digest** Toll-like receptors (TLRs) are proteins that are responsible for recognizing specific molecules associated with invading pathogens, known as pathogen-associated molecular patterns. Upon detecting these signals, TLRs activate the body's immune response, which fights the infection.

A subset of TLRs recognizes nucleic acids, including DNA and RNA, enabling the immune system to respond to foreign material from a diverse range of bacteria and viruses. However, some of the body's own DNA and RNA is also found outside cells (e.g., in the bloodstream) and TLRs must be able to discriminate between these nucleic acids and those belonging to pathogens, because failure to tell the difference between the two could result in autoimmune disease. To reduce this risk, TLRs are sequestered inside the cell within membrane-bound compartments known as endosomes.

UNC93B1 is a transmembrane protein that is known to control the movement of TLRs from the endoplasmic reticulum—where TLRs are assembled—to endosomes. However, the exact mechanisms by which this protein controls TLR trafficking were unclear. Now Lee et al. reveal that it directly controls the packaging of at least six TLRs at the endoplasmic reticulum: it helps to load these TLRs into vesicles, which are in turn processed by the Golgi apparatus—the organelle wherein proteins are sorted and packaged en route to their final destinations. Surprisingly, UNC93B1 remains associated with the TLRs even after Golgi processing.

Lee et al. also reveal that specific endosomal TLRs are subject to distinct post-Golgi trafficking mechanisms. In order for TLR9 to be delivered to the endosome, UNC93B1 must recruit an adaptor protein called AP-2, whereas other TLRs appear to require different actions by UNC93B1. By defining the mechanisms that underlie the differential trafficking of endosomal TLRs, Lee et al. suggest that we may learn how to manipulate distinct aspects of TLR activation, and also gain insights into the causes of certain autoimmune diseases.

TLR9 and other intracellular TLRs must traffic from the endoplasmic reticulum (ER) to endolysosomes before responding to ligands. UNC93B1, a multi-pass transmembrane protein localized to the ER, appears to facilitate this trafficking (*Brinkmann et al., 2007*; *Kim et al., 2008*). Mice homozygous for a nonfunctional Unc93b1 (H412R) allele (*Unc93b1*$^{3d/3d}$) fail to respond to TLR3, TLR7, or TLR9 ligands, and mice and humans deficient in UNC93B1 are highly susceptible to viral infection (*Casrouge et al., 2006*; *Tabeta et al., 2006*; *Lafaille et al., 2012*). More recently, UNC93B1 has been implicated in the function of TLR11, TLR12, and TLR13 (*Pifer et al., 2011*; *Shi et al., 2011*; *Koblansky et al., 2012*; *Oldenburg et al., 2012*). UNC93B1 is not required for responses by surface localized TLRs such as TLR2 and TLR4 (*Tabeta et al., 2006*). UNC93B1 associates with endosomal TLRs, and in cells with defective UNC93B1, TLR9 and TLR7 fail to leave the ER (*Brinkmann et al., 2007*; *Kim et al., 2008*). However, the mechanism by which UNC93B1 facilitates TLR trafficking to endosomal compartments remains enigmatic, especially considering its reported direct translocation from the ER to endolysosomes (*Kim et al., 2008*). This pathway is inconsistent with our findings that TLR9 and TLR7 traffic through the general secretory pathway en route to endosomes (*Ewald et al., 2008*, *2011*). In addition, mice expressing an aspartic acid to alanine mutation at amino acid position 34 in UNC93B1 (*Unc93b1*$^{D34A/D34A}$) were recently shown to develop spontaneous autoimmunity due to enhanced TLR7 responses and diminished TLR9 responses (*Fukui et al., 2009*, *2011*). These findings suggest that regulation of TLRs by UNC93B1 can influence the relative thresholds of receptor activation. For these reasons, we have sought to define the molecular basis by which UNC93B1 controls endosomal TLR trafficking and function.

Beyond the implied role for UNC93B1 discussed above, little is known about the molecular mechanisms that mediate proper localization of endosomal TLRs and no other factor required specifically for endosomal TLR trafficking has been identified. Nevertheless, several reports suggest that endosomal TLR trafficking may be influenced at both ER and post-Golgi trafficking steps. Gp96 functions as an ER folding chaperone for many TLRs, including TLR9, and PRAT4A has been implicated in TLR trafficking from the ER (*Takahashi et al., 2007*; *Yang et al., 2007*; *Lee et al., 2012*). Additionally, the HRS/ESCRT pathway is involved in post-Golgi trafficking by sorting ubiquitinated TLR7 and TLR9 to endosomal compartments (*Chiang et al., 2012*), and the adaptor protein-3 (AP-3) has been reported to target TLR9 and TLR7 to lysosome related organelles specialized for type I IFN induction (*Honda et al., 2005*;

*Blasius et al., 2010*; *Sasai et al., 2010*). Interestingly, UNC93B1 trafficking to these compartments is also impaired in AP-3 deficient cells (*Sasai et al., 2010*). Whether UNC93B1 interacts with other components implicated in trafficking of endosomal TLRs remains to be determined.

In this study, we report that UNC93B1 is required for multiple steps of TLR trafficking. UNC93B1 plays a direct role in facilitating exit of TLRs from the ER as well as a later role in recruitment of adaptor protein-2 (AP-2) to facilitate endocytosis of TLR9 from the plasma membrane. Surprisingly, TLR7 does not have the same requirements for UNC93B1 and utilizes distinct trafficking machinery to reach endolysosomes. Thus, our results describe how UNC93B1 controls endosomal TLR trafficking and provide the first mechanistic basis for differential regulation of these receptors.

## Results

### UNC93B1 traffics to phagosomes via the Golgi compartment

UNC93B1 has been described as an ER-resident trafficking chaperone that translocates TLRs directly from the ER to endolysosomes upon TLR activation (*Kim et al., 2008*). This model is based in part on the observation that UNC93B1 never acquires Endoglycosidase H (EndoH)-resistant glycans (*Brinkmann et al., 2007*), which are acquired only when proteins traffic through the medial Golgi. Because this proposed function for UNC93B1 conflicts with our model of TLR9 trafficking (*Ewald et al., 2008*, *2011*), we first examined whether UNC93B1 is present in endolysosomal compartments in unstimulated cells. Wildtype (WT) UNC93B1 but not the nonfunctional (H412R) mutant was detectable in phagosomes purified from unstimulated RAW264 cells (*Figure 1A*). Moreover, a portion of UNC93B1-WT gained EndoH-resistance in multiple cell types, while UNC93B1-H412R was entirely EndoH-sensitive (*Figure 1B–F*). These results agree with a previous report that the H412R mutant fails to leave the ER (*Kim et al., 2008*). To formally demonstrate that the increased molecular weight of UNC93B1-WT is due to N-linked glycans, we mutated Asn-251, which is within a consensus N-glycosylation site, and this mutant failed to acquire EndoH-resistant glycans (*Figure 1G*). Based on the acquisition of EndoH-resistant glycans by UNC93B1, we examined whether UNC93B1 is detectable within COPII vesicles, which mediate transport of cargo between the ER and Golgi (*Zanetti et al., 2012*). Using an in vitro COPII budding assay (*Kim et al., 2005*; *Merte et al., 2010*), we compared levels of UNC93B1-WT and UNC93B1-H412R in purified vesicles. UNC93B1-WT, but not H412R, was clearly present within the vesicles, further supporting a model in which UNC93B1 exits the ER through the general secretory pathway (*Figure 1H*). Altogether, these data indicate that a pool of UNC93B1 protein exits the ER and traffics through the Golgi in unstimulated cells. Moreover, transit from the ER to the Golgi may be important for UNC93B1 function, as the nonfunctional UNC93B1-H412R mutant fails to enter COPII vesicles and does not reach the medial Golgi.

### UNC93B1 facilitates TLR9 loading into COPII vesicles

Our previous work reported that three species of TLR9 can be detected within macrophages, representing distinct maturation stages: an initial 150-kDa species with EndoH-sensitive glycans corresponding to the ER-resident protein (TLR9-ER), a larger species with EndoH-resistant glycans corresponding to full-length receptor that has passed through the Golgi (TLR9-Precursor), and a 80-kDa band with EndoH-resistant glycans corresponding to the mature, cleaved receptor within endolysosomes (TLR9-Cleaved) (*Figure 2A*, lane 1) (*Ewald et al., 2008*, *2011*). To examine how UNC93B1 function impacts TLR9 localization, we compared these three forms of TLR9 in immortalized macrophages derived from *Unc93b1*[3d/3d] mice and complemented with UNC93B1-WT or UNC93B1-H412R. While all three bands were present in macrophages with functional UNC93B1, only the ER-resident form was detectable in cells expressing the UNC93B1-H412R mutant (*Figure 2A*), consistent with our previous analysis of TLR9 in UNC93B1 shRNA knockdown cells (*Ewald et al., 2008*). These data indicate that TLR9 does not reach the medial Golgi in the absence of functional UNC93B1. Because a pool of UNC93B1 can traffic from ER to Golgi by entering COPII vesicles (*Figure 1B*), we considered whether UNC93B1 regulates this aspect of TLR9 trafficking. Indeed, analysis of TLR9 loading into COPII vesicles revealed that TLR9 was only detectable in the presence of functional UNC93B1 whereas a control traffic protein, ERGIC/p58, was packaged independently of UNC93B1 (*Figure 2B*).

Quality control mechanisms ensure that only properly folded proteins can exit the ER, and one mechanism by which UNC93B1 could regulate ER exit of TLR9 is through regulation of TLR9 folding, as has been reported for gp96 (*Yang et al., 2007*). To address whether UNC93B1 serves as a folding

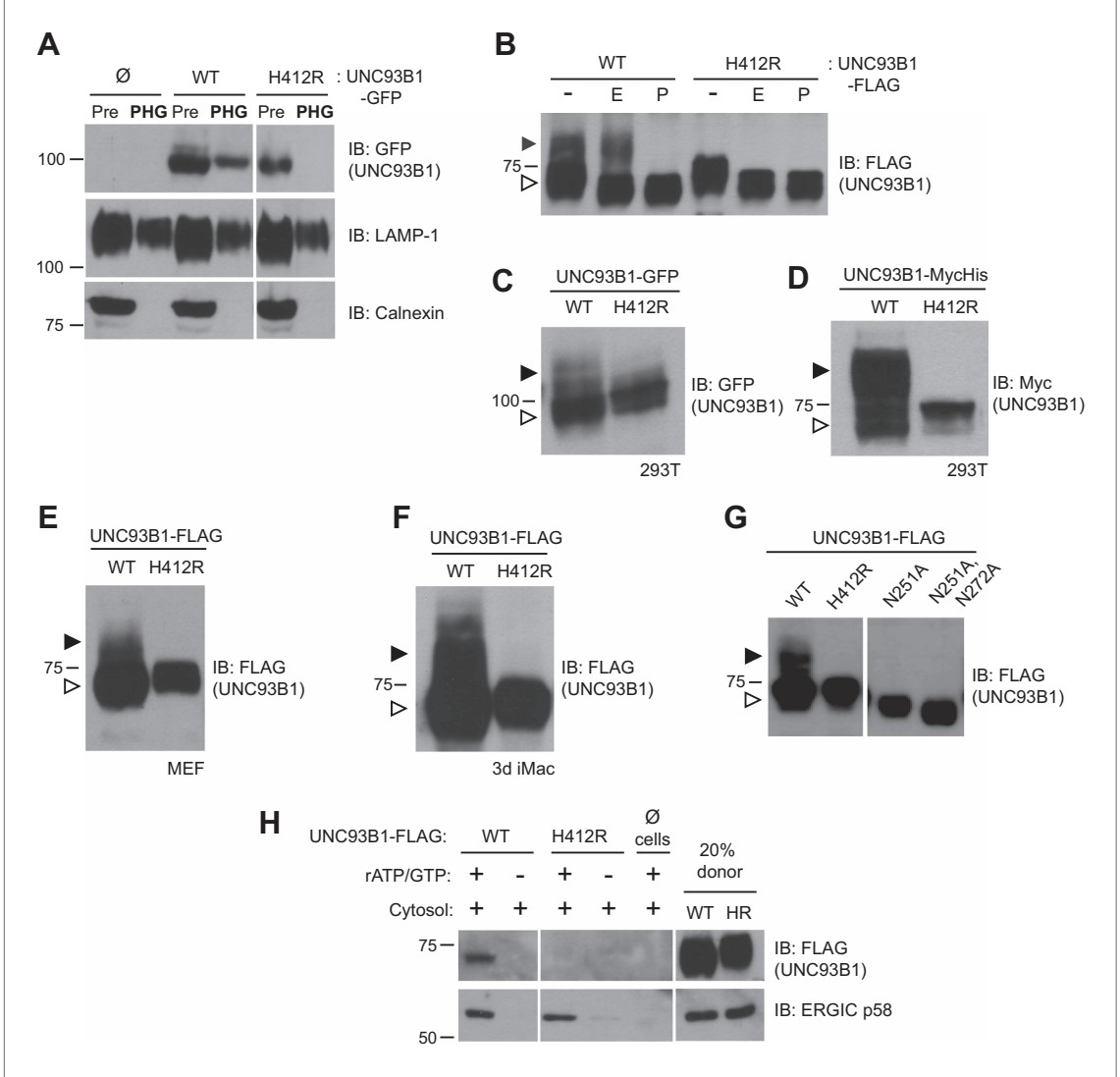

**Figure 1**. UNC93B1 traffics to the Golgi en route to endolysosomes. (**A**) UNC93B1 is present in phagolysosomes of unstimulated cells. Phagosomes (PHG) isolated by flotation from RAW264 cells only (Ø) or expressing GFP tagged UNC93B1-WT or UNC93B1-H412R and cells prior to isolation (Pre) were separated by SDS-PAGE, and immunoblotted with anti-GFP, anti-LAMP1 (lysosome marker), and anti-calnexin (ER marker). (**B**) A portion of UNC93B1 protein traffics to the Golgi apparatus. Wildtype UNC93B1 (WT) or H412R, each with a C-terminal 3× FLAG tag, were expressed in HEK293Ts by transient transfection. The immunoprecipitated proteins were treated with EndoH (E), PNGaseF (P) or left untreated (−), separated by SDS-PAGE, and visualized by immunoblot with anti-FLAG antibody. Bands representing EndoH-sensitive (white arrow) and resistant (black arrow) forms of UNC93B1 are indicated. (**C**)–(**F**) UNC93B1 acquires EndoH-resistant modifications. UNC93B1 tagged with GFP (**C**) or myc-His (**D**) from transiently transfected HEK293Ts, and FLAG tagged UNC93B1 expressed in MEFs (**E**) or *3d* iMac cells (**F**) were analyzed for the presence of EndoH-resistant glycans. Lysates were separated by SDS-PAGE and immunoblotted with the indicated antibodies. EndoH-sensitive (white arrow) and EndoH-resistant (black arrow) forms are indicated. (**G**) Mutation of UNC93B1 glycosylation sites abolishes EndoH resistant forms. Lysates from HEK293Ts transiently transfected with FLAG tagged UNC93B1-WT, -N251A or -N251A/N272A were separated by SDS-PAGE and immunoblotted with anti-FLAG antibody. (**H**) UNC93B1 is loaded into COPII vesicles. Digitonin-permeabilized COS7 cells expressing 3× FLAG-tagged UNC93B1-WT or UNC93B1-H412R, or no cells (Ø) were incubated with ATP regenerating system, GTP, and rat liver cytosol, as indicated, in an in vitro COPII budding assay. Vesicles purified by ultracentrifugation were analyzed by SDS-PAGE and immunoblot using the indicated antibodies. 20% of the COS7 cells prior to the budding reaction serves as a loading control (20% donor). ERGIC/p58 serves as a positive control for the formation of COPII vesicles. Results are representative of at least three experiments (**A**–**G**) or two experiments (**H**).

chaperone, we tested whether a chimeric CD4-TLR9 protein, consisting of the ectodomain of CD4 fused to the transmembrane and cytosolic regions of TLR9 (*Figure 2C*, left), required UNC93B1 function. Because trafficking of CD4 is not UNC93B1-dependent (*Figure 2D*), this chimera can be used to test whether TLR9 requires UNC93B1 to ensure correct folding of the TLR9 ectodomain. CD4-TLR9

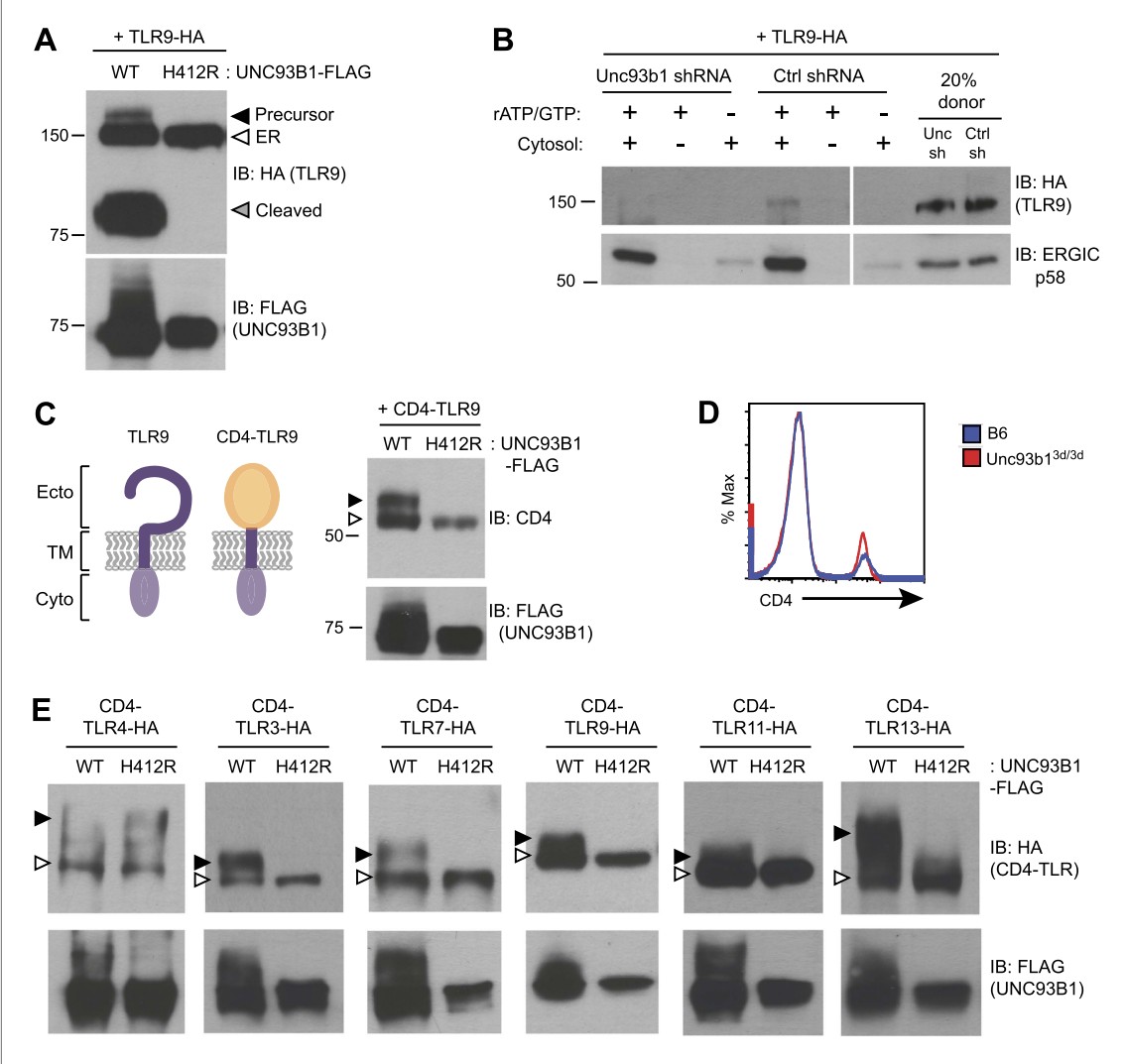

**Figure 2**. UNC93B1 controls ER exit of TLRs 3, 7, 9, 11, and 13. (**A**) TLR9 fails to exit the ER in cells lacking functional UNC93B1. Lysates from *3d* iMac cells complemented with either UNC93B1-WT or UNC93B1-H412R and expressing TLR9-HA were analyzed by SDS-PAGE and immunoblotted with the indicated antibodies. The precursor (black arrow), ER (white arrow) and cleaved (grey arrow) forms of TLR9-HA are indicated. (**B**) UNC93B1 is required for TLR9 loading into COPII vesicles. RAW264 macrophages stably transduced with retroviruses encoding control or Unc93b1-directed shRNA and expressing TLR9-HA were used in an in vitro COPII budding assay as described in (**Figure 1H**). Lysates of purified vesicles or donor membranes were probed with the indicated antibodies. (**C**) The transmembrane and cytosolic domain of TLR9 is sufficient to confer UNC93B1-dependence. (Left) schematic of TLR9 and the CD4-TLR9 chimera. Transmembrane (TM), ectodomain (Ecto) and cytosolic domain (Cyto) are indicated. (Right) CD4-TLR9 was expressed in HEK293Ts together with FLAG-tagged UNC93B1-WT or UNC93B1-H412R. Total lysates were analyzed by SDS-PAGE and immunoblotted with anti-CD4 and anti-FLAG antibodies. EndoH-sensitive (white arrow) and resistant (black arrow) forms are indicated. (**D**) CD4 trafficking to the cell surface is normal in *Unc93b1*[3d/3d] cells. Splenocytes from C57BL/6 (blue line) or *Unc93b1*[3d/3d] (red line) mice were stained with anti-CD4 and analyzed by flow cytometry. (**E**) CD4-TLR chimeric proteins for each of the indicated TLRs were expressed in HEK293Ts together with FLAG-tagged UNC93B1-WT or UNC93B1-H412R. Lysates were separated by SDS-PAGE and visualized by immunoblot with anti-HA and anti-FLAG antibodies. EndoH-sensitive (white arrows) and resistant (black arrows) forms are indicated. The chimeras were constructed as shown in **Figure 1E**, except with the addition of a C-terminal HA tag. Results are representative of at least three experiments (**A**, **C**, and **E**) or two experiments (**B** and **D**).

acquired EndoH-resistant glycans when expressed with UNC93B1-WT but not when expressed with mutant UNC93B1-H412R (**Figure 2C**, right). Thus, the requirement for UNC93B1 is not based on TLR9 ectodomain folding. Furthermore, the transmembrane domain and cytosolic regions of TLR9 are sufficient to mediate UNC93B1-dependent trafficking.

Taken together, these data indicate that UNC93B1 regulates ER to Golgi transport of TLR9. While we cannot rule out that a pool of UNC93B1 bypasses the Golgi en route to endosomes as suggested

by others (*Brinkmann et al., 2007*; *Kim et al., 2008*), this route does not seem relevant for UNC93B1-dependent TLR9 trafficking. Instead, UNC93B1 appears to control TLR9 entry into COPII vesicles.

## UNC93B1 is required for proper trafficking of TLR3, TLR7, TLR9, TLR11, and TLR13 from the ER

We next sought to determine whether the dependence on UNC93B1 for ER exit is a general property of endosomal TLRs. We generated a panel of CD4-TLR fusion proteins and tested whether they required UNC93B1 to exit the ER, based on acquisition of EndoH-resistant glycans. As expected, ER exit of both CD4-TLR3 and CD4-TLR7 required UNC93B1, which is consistent with defective TLR3 and TLR7 signaling in *Unc93b1*[3d/3d] mice (*Tabeta et al., 2006*), whereas CD4-TLR4 trafficked independently of UNC93B1 (*Figure 2E*). CD4-TLR11 and CD4-TLR13 also required UNC93B1 for ER exit (*Figure 2E*). Less is known about the localization or trafficking of these TLRs, although biochemical studies have suggested that these TLRs can associate with UNC93B1 (*Brinkmann et al., 2007*; *Melo et al., 2010*; *Pifer et al., 2011*). Our results are the first to show that UNC93B1 regulates the trafficking of TLR11 and TLR13 by controlling ER exit. Altogether, our data indicate that the role for UNC93B1 in controlling ER exit extends to all known endosomal TLRs (TLR3, TLR7, TLR9) as well as TLR11 and TLR13.

## Mutational analysis reveals distinct regions of UNC93B1 required for ER exit and post-Golgi trafficking of TLR9

To define regions within UNC93B1 necessary for ER export of TLRs, we generated a series of N-and C-terminal truncation mutants. Based on the predicted topology of UNC93B1 (*Brinkmann et al., 2007*), the N- and C-termini face the cytosol, so we reasoned that these regions would be most likely to interact with putative trafficking factors (*Figure 3A*). Interestingly, N-terminal truncations resulted in a strong defect in TLR9 trafficking from the ER as evidenced by significantly reduced or absent precursor and cleaved forms of TLR9 in cells expressing the Δ50 and Δ57 UNC93B1 truncations (*Figure 3B*). This defective trafficking resulted in impaired responses to TLR9 ligands in macrophages, while TLR2 responses remained intact (*Figure 3C*). Taken together, our data indicate that the N-terminal region of UNC93B1 contains residues important for trafficking of TLR9 and UNC93B1 from the ER to endolysosomes. These findings are in agreement with a previous study showing that TLR9 responsiveness is strongly dependent on the N-terminus of UNC93B1, although this study did not directly examine TLR9 trafficking (*Fukui et al., 2009*). Furthermore, an UNC93B1 point mutant (D34A), which reduced TLR9 signaling but enhanced TLR7 signaling (*Fukui et al., 2009, 2011*), reduced TLR9 transport and cleavage (*Figure 3D*).

Our results thus far suggest that UNC93B1 controls the function of multiple TLRs by regulating their exit from the ER. However, UNC93B1 itself also exits the ER and is present in endolysosomes (*Figure 1A,H*). Moreover, we could detect both the precursor and cleaved forms of TLR9 associated with immunoprecipitated UNC93B1, suggesting that UNC93B1 and TLR9 remain associated after leaving the ER (*Figure 3E*). To test whether this post-ER interaction may have functional consequences, we screened UNC93B1 mutants for any role in TLR9 function beyond ER export. Strikingly, truncation of the UNC93B1 C-terminus (Δ523 and Δ538) resulted in an accumulation of the Golgi-modified precursor form of TLR9 and a marked reduction of the cleaved receptor (*Figure 3B*). This pattern suggests that TLR9 trafficking is blocked at some point after the medial Golgi, in contrast to the trafficking defect observed with N-terminal deletion mutants. Consistent with this interpretation, we observed reduced TLR9 signaling in cells expressing the Δ523 and Δ538 UNC93B1 mutants (*Figure 3C*). Thus, in addition to its role in ER export, UNC93B1 appears to regulate post-ER trafficking of TLR9.

## UNC93B1 controls post-Golgi trafficking of TLR9 through recruitment of AP-2

We next sought to define the mechanism underlying the requirement for UNC93B1 in post-ER TLR9 trafficking. We focused on residues 539–542, YRYL, because they are evolutionarily conserved and fit the consensus for a YxxΦ motif (where Φ represents a hydrophobic residue), which may mediate protein interactions or serve as a site for phosphorylation (*Songyang et al., 1993*; *Ohno et al., 1995, 1996*; *Crump et al., 1998*; *Bonifacino and Traub, 2003*) (*Figure 4A*). Mutation of Tyr539 to an Ala (Y539A) resulted in accumulation of the Golgi-modified precursor form of TLR9 and reduced TLR9 cleavage, similar to the block in TLR9 trafficking observed with the Δ538 truncation mutant (*Figure 4B*). Mutation of Leu-542 to Ala (L542A) and Tyr-539 to Phe (Y539F) gave similar results, although the block was not as complete as Y539A (*Figure 4C*). Responsiveness to TLR9 ligands was also reduced in UNC93B1-Y539A expressing cells, consistent with the block of TLR9 processing (*Figure 4D*).

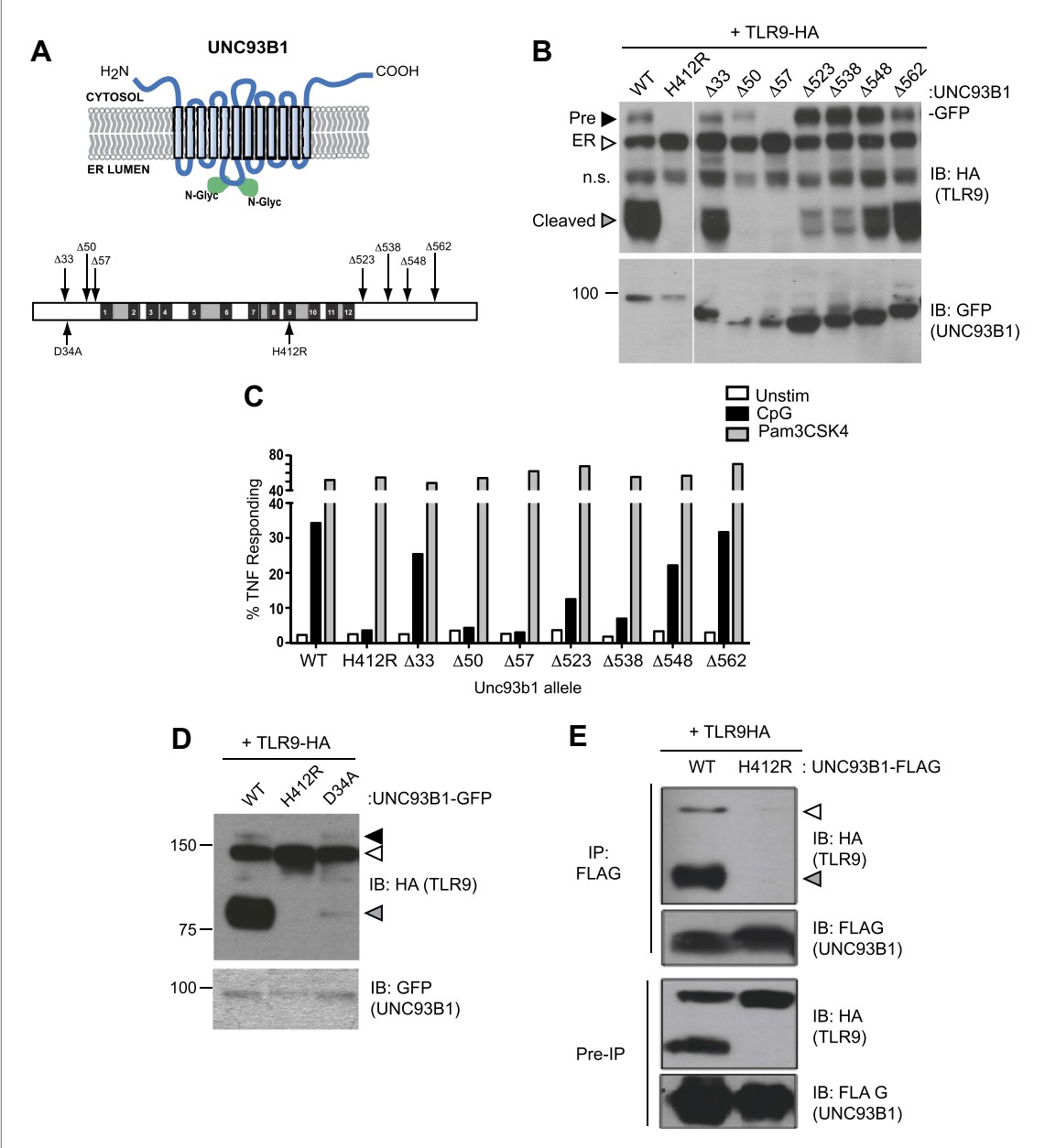

**Figure 3**. UNC93B1 mutants reveal two distinct roles in TLR9 trafficking. (**A**) Schematic of predicted UNC93B1 topology. UNC93B1 is a 12 pass trans-membrane protein with terminal regions facing cytosol. Two putative N-linked glycosylation (N-Glyc) sites are indicated (top). Schematic of UNC93B1 (1–598 a.a.). The black numbered boxes represent predicted transmembrane domains, white boxes represent regions predicted to face the cytosol, and grey boxes represent regions predicted to face the lumen. Truncation and point mutations are indicated with arrows (bottom). (**B**) N- and C-terminal mutants of UNC93B1 have distinct TLR9 trafficking outcomes. Lysates from *3d* iMac cells expressing TLR9-HA and complemented with mutant forms of GFP-tagged UNC93B1 were subjected to SDS-PAGE and immunoblotted with anti-HA and anti-GFP antibodies. The precursor (black arrow), ER (white arrow) and cleaved (grey arrow) forms of TLR9-HA are indicated. n.s. indicates a non-specific band. (**C**) N- and C-terminal mutants of UNC93B1 have diminished TLR9 signaling. *3d* iMac cells complemented with WT or indicated mutant alleles of UNC93B1 were harvested for intracellular TNFα staining 5 hr after stimulation with 3 μM CpG, 1 μg/ml Pam3CSK4 or left unstimulated. Percentages of TNF-producing cells after gating on UNC93B1-GFP positive cells are plotted. (**D**) TLR9 trafficking to endolysosomes is impaired in UNC93B1-D34A cells. *3d* iMac cells were complemented with GFP-tagged UNC93B1-WT, -H412R or -D34A. Lysates were separated by SDS-PAGE and visualized by immunoblot with anti-HA and anti-GFP antibodies. The precursor (black arrow), ER (white arrow) and cleaved (grey arrow) forms of TLR9-HA are indicated. (**E**) UNC93B1 interacts with the cleaved form of TLR9. Immunoprecipitation of FLAG tagged UNC93B1 (WT or H412R) in *3d* iMac cells expressing TLR9-HA was performed in 1% digitonin with anti-FLAG matrix. UNC93B1 associated proteins were analyzed by SDS-PAGE and immunoblot with anti-FLAG and anti-HA antibodies. ER (white arrows) and cleaved (grey arrows) forms of TLR9-HA are indicated. All results are representative of at least three experiments.

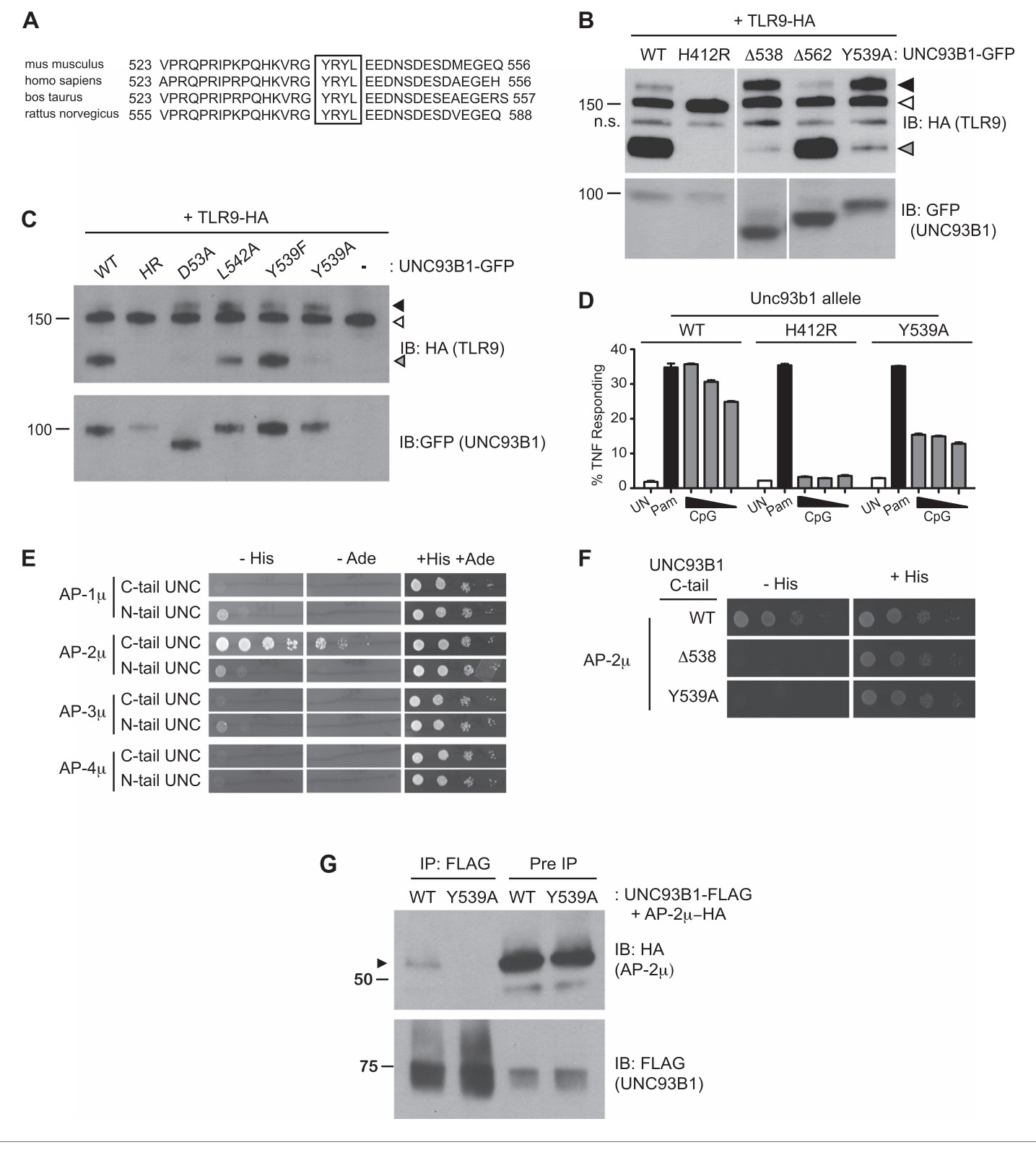

**Figure 4**. UNC93B1 controls post-Golgi trafficking of TLR9 by recruiting AP-2. (**A**) Multi-species protein sequence alignment of residues in the UNC93B1 C-terminal tail. The YxxΦ motif, YRYL, is boxed. (**B**) Precursor TLR9 accumulates in UNC93B1-Y539A expressing cells. Lysates from *3d* iMac cells expressing TLR9-HA were complemented with the indicated GFP tagged UNC93B1, analyzed by SDS-PAGE, and immunoblotted with anti-HA and anti-GFP antibodies. The precursor (black arrow), ER (white arrow) and cleaved (grey arrow) forms of TLR9-HA are indicated. n.s. indicates a non-specific

*Figure 4. Continued on next page*

*Figure 4. Continued*

band. (**C**) Mutations of YxxΦ in UNC93B1 confer partial phenotypes in TLR9 trafficking when compared with Y539A. Lysates from *3d* iMac cells expressing TLR9-HA and GFP tagged UNC93B1-WT, H412R, Δ538, L542A, Y539F, Y539A were analyzed by SDS-PAGE and immunoblotted with anti-HA and anti-GFP antibodies. Presence of precursor (black arrow), ER (white arrow) and cleaved (grey arrow) forms of TLR9 are indicated. (**D**) TLR9 signaling is impaired in UNC93B1-Y539A cells. *3d* iMac cells, complemented with GFP tagged UNC93B1-WT, -H412R, or -Y539A, were harvested for intracellular TNFα staining 5 hr after stimulation with 3 μM CpG, 1 μg/ml Pam3CSK4 (Pam) or left unstimulated. Percentages of TNF-producing cells after gating on UNC93B1-GFP positive cells are plotted. (**E**) The C-terminal tail of UNC93B1 interacts with AP-2μ. Results from a yeast two-hybrid assay testing for interaction between the AP (-1A, -2, -3A, -4) μ subunits and the N- or C-terminal cytosolic regions of UNC93B1 (N- or C-tail UNC), are shown. Growth on –His–Trp–Leu plates (–His) or –Ade–Trp–Leu (–Ade) indicates interaction. Growth on –Trp–Leu plates (+His+Ade) serves as a control. (**F**)–(**G**) Tyr-539 on UNC93B1 mediates interaction with AP-2 complex. (**F**) Results from a yeast two-hybrid assay testing for interaction between the AP-2μ subunit and the C-terminal cytosolic region of UNC93B1 (UNC93B1 C-tail) from WT, the Y539A mutant, or the Δ538 mutant are shown. Growth on –His–Trp–Leu plates (–His) indicates interaction. Growth on –Trp–Leu plates (+His) serves as a control. (**G**) HEK293T were transiently transfected with AP-2μ-HA and FLAG tagged UNC93B1-WT, -Y539A. Cell lysates were incubated with anti-FLAG matrix, and UNC93B1-associated proteins were eluted with FLAG peptide, separated by SDS-PAGE and visualized by immunoblot with anti-HA or anti-FLAG antibodies. UNC93B1 associated AP-2μ is indicated with black arrow. Results are representative of at least three experiments (**B**, **D**–**F**) or two experiments (**C** and **G**).

YxxΦ motifs can serve as binding sites for clathrin adaptor protein (AP) complexes, so we tested whether the C-terminal tail of UNC93B1 could interact with any of the four mammalian AP complexes. AP complexes consist of four subunits, of which the μ subunit typically determines cargo specificity (***Bonifacino and Traub, 2003***; ***Ohno, 2006***). Because these interactions are often weak, we used a yeast two-hybrid (Y2H) assay in which the μ subunits of AP-1, AP-2, AP-3, and AP-4 were fused to the Gal4 activation domain and the UNC93B1 N- and C-terminal tails were fused to the Gal4 DNA binding domain. The C-terminal cytosolic region of UNC93B1 interacted strongly with AP-2μ, but not with the μ subunits of any of the other AP complexes (***Figure 4E***). Importantly, interaction with AP-2μ was abolished by the Δ538 truncation and the Y539A mutation (***Figure 4F***). Additionally, we were able to co-immunoprecipitate AP-2μ with UNC93B1-WT, but not with UNC93B1-Y539A (***Figure 4G***). Taken together, these data suggest that UNC93B1 directly recruits AP-2 via a YxxΦ motif present in its C-terminal cytosolic tail.

AP-2 complexes direct clathrin-mediated endocytosis of cargo from the plasma membrane. Therefore, our results suggest that UNC93B1 and TLR9 traffic together to the surface and require AP-2-mediated internalization to reach endocytic compartments. To test this possibility, we examined whether TLR9 required UNC93B1-mediated recruitment of AP-2 to gain access to endolysosomal compartments. Using immunofluorescence microscopy, we detected reduced colocalization of TLR9 and the lysosomal marker Lamp-1 in UNC93B1-Y539A-expressing cells (***Figure 5A***). We also compared surface expression of N-terminally tagged TLR9 in cells expressing different UNC93B1 alleles. While surface TLR9 was only weakly detectable in cells expressing UNC93B1-WT, expression of UNC93B1-Y539A increased the levels of FLAG-TLR9 at the cell surface (***Figure 5B***). Furthermore, as we found for TLR9-HA, the FLAG-TLR9 Golgi-modified precursor form accumulated in UNC93B1-Y539A expressing cells (***Figure 5C***). This result suggests that disruption of the interaction between UNC93B1 and AP-2 leads to defective endocytosis of TLR9 from the cell surface. In support of this model, siRNA knockdown of AP-2μ1 resulted in a similar increase in surface expression of TLR9 (***Figure 5D***) and accumulation of the TLR9 precursor form (***Figure 5E***). As a positive control for AP-2μ1 knockdown, we observed accumulation of CD71 (transferrin receptor) at the cell surface (***Figure 5D***). Taken together, these data support a model in which UNC93B1 interacts with AP-2 to traffic TLR9 from the cell surface to endolysosomes.

## UNC93B1 differentially controls TLR7 and TLR9

Having established the mechanisms by which UNC93B1 facilitates TLR9 delivery to endolysosomes, we next examined whether TLR7 is under similar regulation. However, for reasons that remain unclear, detection of endogenous or epitope-tagged exogenous TLR7 protein is quite challenging. Consequently, the evidence that TLR7 undergoes ectodomain processing is limited (***Ewald et al., 2011***), and one group has reported that TLR7 is not cleaved (***Park et al., 2008***). We synthesized a TLR7 coding sequence optimized for efficient translation (see 'Materials and methods') and found that our ability to detect TLR7 protein was significantly improved. Cleaved TLR7 was detected in macrophages expressing UNC93B1-WT but not in macrophages expressing UNC93B1-H412R (***Figure 6A***). Both the full-length and cleaved forms of TLR7 possessed EndoH-resistant glycans, although fewer of the

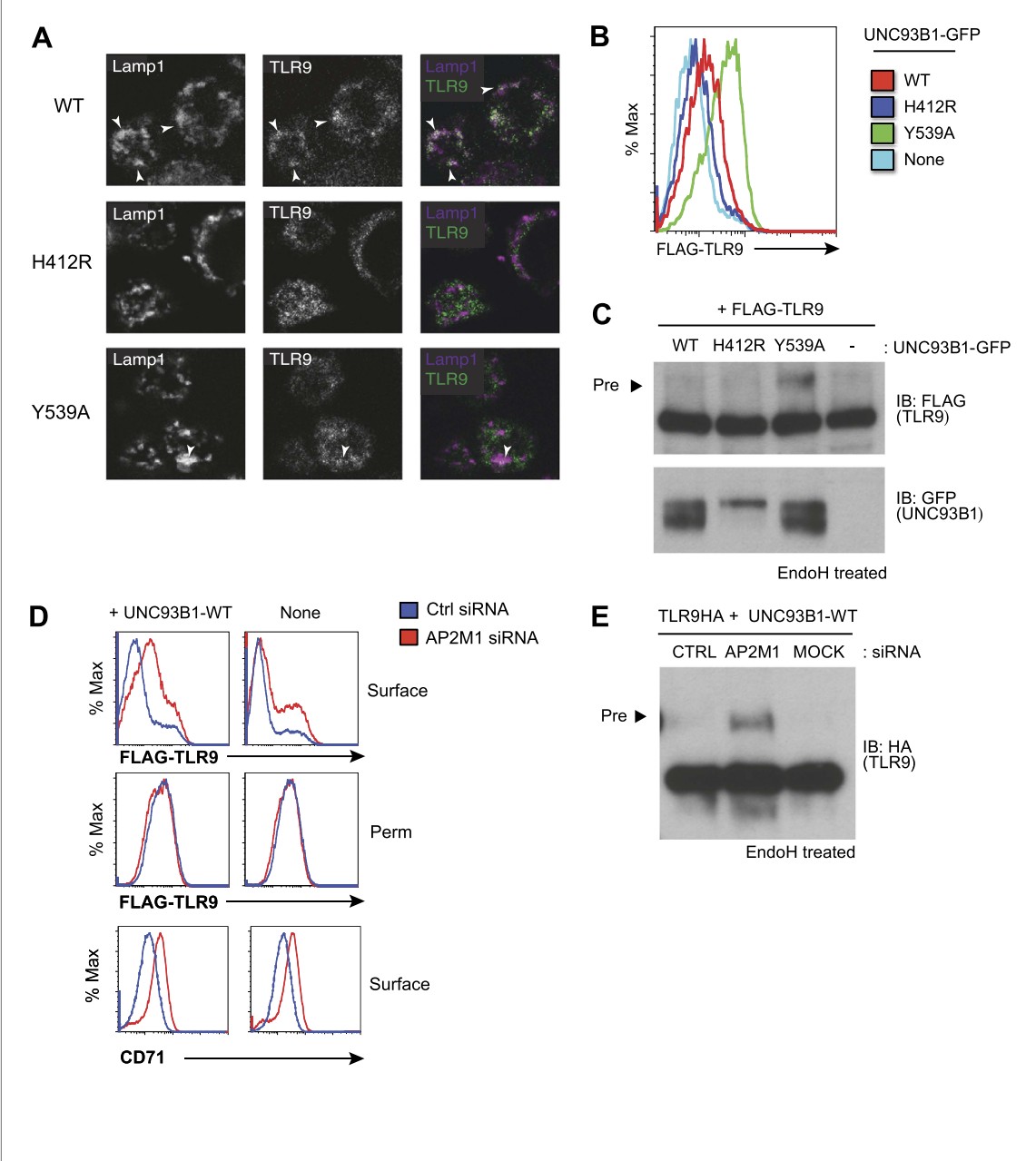

**Figure 5**. Failure to recruit AP-2 by UNC93B1 results in cell surface accumulation of TLR9. (**A**) TLR9 fails to traffic to endolysosomes in UNC93B1-Y539A expressing cells. Localization of TLR9-HA and Lamp-1 in *3d* iMac cells complemented with UNC93B1-WT, -H412R, or -Y539A was determined by immuno-fluorescence microscopy. Representative images of Lamp-1 (left), TLR9 (middle), and a pseudo-colored merged image for each UNC93B1 allele are shown. Arrowheads indicate areas of Lamp-1/TLR9 colocalization. (**B**) TLR9 accumulates at the cell surface in UNC93B1-Y539A expressing cells. HEK293Ts transfected with N-terminally tagged 3× FLAG-TLR9 and the indicated UNC93B1 alleles were stained with anti-FLAG and goat anti-mouse IgG secondary antibodies. FLAG staining was measured by flow cytometry. (**C**) The TLR9 precursor form accumulates in HEK293Ts expressing N-terminally tagged FLAG-TLR9 and UNC93B1-Y539A. Lysates from HEK293Ts stably expressing N-terminally tagged 3× FLAG TLR9 and GFP tagged UNC93B1-WT, -H412R, or -Y539A were harvested, treated with EndoH, separated by SDS-PAGE, and visualized by immunoblot with anti-FLAG and anti-GFP antibodies. (**D**) TLR9 accumulates at the cell surface in cells lacking AP-2. HEK293Ts stably expressing 3× FLAG-TLR9 with or without WT UNC93B1 were treated with AP-2μ (AP2M1) or control siRNA for 96 hr. FLAG staining was measured on intact (surface) or permeabilized (Perm) cells as described in (**G**). Anti-CD71 (transferrin receptor) staining serves as a control for AP-2 knockdown. (**E**) Absence of AP-2 causes accumulation of TLR9 precursor form. HEK293Ts were treated with control or AP-2μ siRNA. After 48 hr, siRNA treated cells were transiently transfected with TLR9-HA and GFP tagged UNC93B1-WT. Lysates were harvested 24 hr later for EndoH assay as described in (**A**). The TLR9 precursor form (black arrow) is indicated in each panel. Results are representative of at least three experiments (**B** and **D**) Or two (**A**, **C**, and **E**).

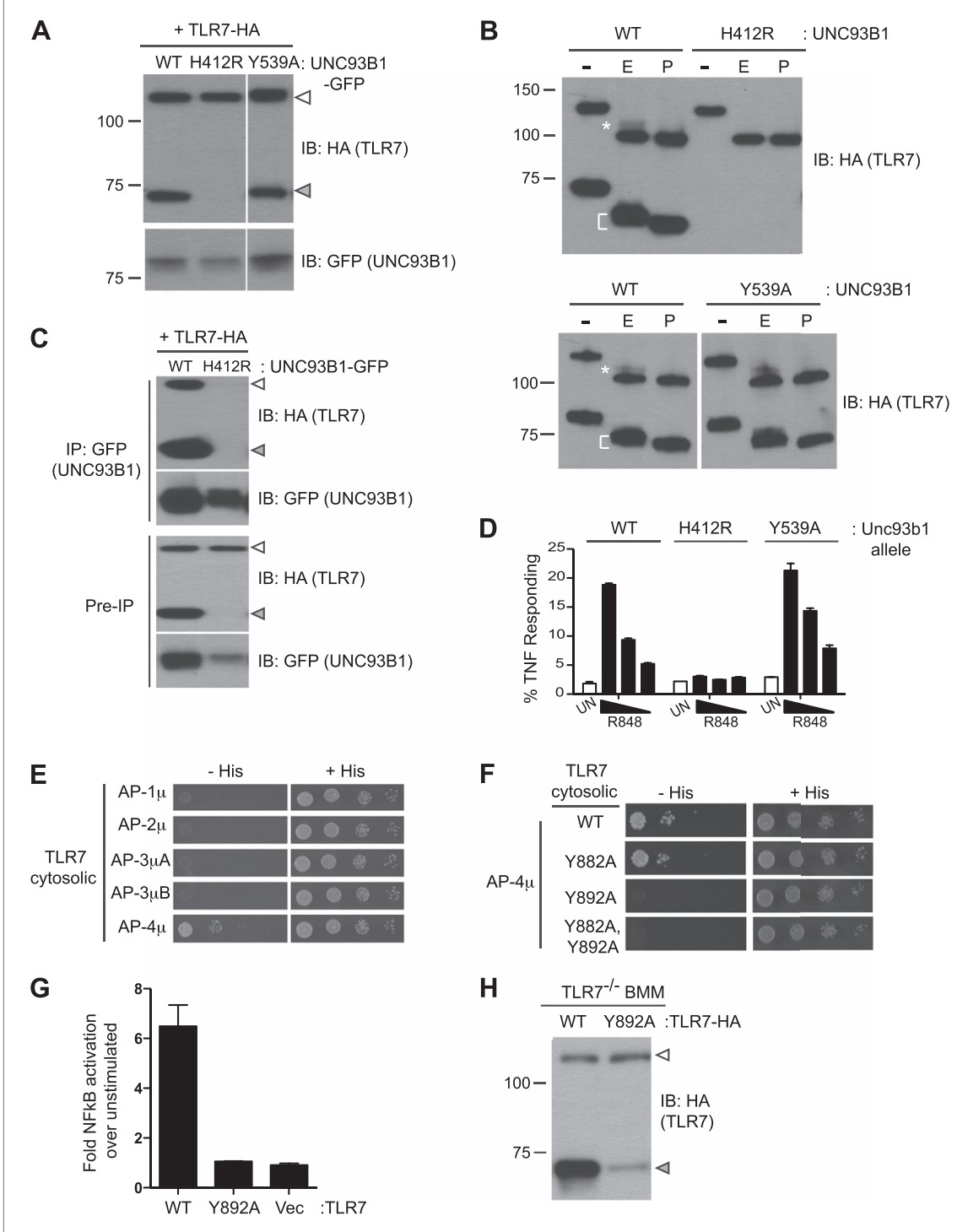

**Figure 6**. Differential trafficking of TLR7 and TLR9. (**A**) TLR7 trafficking can be monitored biochemically and is UNC93B1-dependent. Lysates from *3d* iMac cells expressing TLR7-HA and complemented with GFP-tagged UNC93B1-WT, -H412R or -Y539A were separated by SDS-PAGE and immuno-blotted with anti-HA and anti-GFP antibodies. ER (white arrows) and cleaved (grey arrows) forms of TLR7-HA are indicated. (**B**) TLR7 acquires EndoH resistance. TLR7-HA was immunoprecipitated from *3d* iMac cells expressing UNC93B1-WT, -H412R, or -Y539A, treated with EndoH (E), PNGaseF (P) or left untreated (−), and visualized by anti-HA immunoblot. Asterisk indicates precursor form of the receptor. The bracket indicates the migration difference between EndoH-treated and PNGaseF-treated TLR7. (**C**) UNC93B1 interacts with ER and cleaved forms of TLR7. GFP-tagged UNC93B1-WT or -H412R were immunoprecipitated from *3d* iMac cells expressing TLR7-HA. UNC93B1 associated proteins were analyzed by SDS-PAGE and immuno-blotted with anti-HA and anti-GFP antibodies. ER (white arrows) and cleaved (grey arrows) forms of TLR7-HA are indicated. (**D**) TLR7 signaling is normal in

*Figure 6. Continued on next page*

Figure 6. Continued

UNC93B1-Y539A cells. *3d* iMac cells complemented with GFP tagged UNC93B1-WT, -H412R, or -Y539A were stimulated with 100 to 25 ng/ml R848 for 5 hr and intracellular TNFα stain was performed. Percentages of TNF-producing cells after gating on UNC93B1-GFP positive cells are plotted. (**E**) TLR7 interacts with AP-4μ. Results from a yeast two-hybrid assay testing for interaction between the AP-1, AP-2, AP-3, and AP-4 μ subunits and the C-terminal cytosolic region of TLR7. Growth on –His–Trp–Leu plates (–His) indicates interaction. Growth on –Trp–Leu plates (+His) serves as a control. (**F**) TLR7-Y892A is unable to interact with AP-4μ. Results from a yeast two-hybrid assay testing for interaction between the AP-4μ and TLR7 YxxΦ mutants (Y882A, Y892A, or double). Conditions are as described in (**E**). (**G**) TLR7-Y892A does not respond to TLR7 ligands. HEK293T cells were transiently transfected with an NF-κB luciferase reporter as well as expression plasmids encoding TLR7, TLR7-Y892A, or empty vector. Luciferase production was assayed 16 hr after stimulation with 10 μg/ml R848. (**H**) TLR7-Y892A trafficking is impaired. TLR7$^{-/-}$ bone marrow derived macrophages (BMMs) were transduced with HA-tagged TLR7-WT or TLR7-Y892A. Cell lysates were analyzed by SDS-PAGE and immunoblotted with anti-HA antibodies. ER (white arrows) and cleaved (grey arrows) forms of TLR7-HA are indicated. Results are representative of at least three experiments (**A**, **D**, and **E**) or two experiments (**B**, **C**, and **F**–**H**).

TLR7 N-linked glycans acquired these modifications than for TLR9 (*Figure 6B*, top). These results confirm our previous findings that TLR7 undergoes ectodomain proteolysis and traffics through the Golgi en route to endolysosomes. Importantly, our results confirm that TLR7 trafficking is dependent on UNC93B1.

We next examined whether UNC93B1 controls post-Golgi sorting of TLR7 as it does for TLR9. Similar to TLR9, both the full-length and cleaved forms of TLR7 associated with UNC93B1 suggesting that UNC93B1 traffics with TLR7 to endolysosomes (*Figure 6C*). However, unlike TLR9, the response to TLR7 ligands was normal in UNC93B1-Y539A expressing cells (*Figure 6D*). Moreover, the amount of cleaved TLR7 was unaffected, and the TLR7 Golgi-modified precursor form did not accumulate in these cells (*Figure 6A,B*). Thus, delivery of TLR7 to endolysosomes appears to be independent of the UNC93B1/AP-2 pathway.

If TLR7 does not require the UNC93B1-mediated recruitment of AP-2 for proper trafficking, then how does the receptor reach endolysosomes? We considered the possibility that TLR7 may recruit AP-2 directly, thereby obviating the need for UNC93B1. However, when we screened by Y2H for interactions between the cytosolic region of TLR7 and AP-1, AP-2, AP-3, and AP-4, we found that TLR7 interacted specifically with AP-4μ but not AP-2μ (*Figure 6E*). We identified three potential YxxΦ motifs within TLR7, and mutating one of these (Tyr-892) disrupted the interaction with AP-4 (*Figure 6F*). In addition, a TLR7-Y892A mutant no longer responded to TLR7 ligands (*Figure 6G*) and displayed reduced ectodomain processing when expressed in bone marrow derived macrophages, suggesting that AP-4 is required for TLR7 delivery to endosomes (*Figure 6H*). AP-4 has been implicated in vesicular trafficking between the trans Golgi network (TGN) and endosomes (*Dell'Angelica et al., 1999*; *Aguilar et al., 2001*; *Simmen et al., 2002*; *Barois and Bakke, 2005*; *Burgos et al., 2010*). Therefore, the association between AP-4 and TLR7 suggests that TLR7 is diverted from the secretory pathway at the TGN and delivered directly to endosomes while continually associating with UNC93B1. Thus, the mechanisms controlling trafficking of TLR7 and TLR9 to the compartments from which they signal are distinct.

## Several UNC93B1 dependent TLRs traffic independently of the UNC93B1/AP-2 pathway

We next sought to determine whether the post-Golgi trafficking of other UNC93B1-dependent TLRs is similar to either of the pathways we have described for TLR7 and TLR9. First, we tested whether TLR3, TLR11, and TLR13 responses were affected in UNC93B1-Y539A expressing cells. Unfortunately, both wildtype and UNC93B1-Y539A cells were unresponsive to Poly I:C, profilin, and flagellin (data not shown), so we could not evaluate TLR3 or TLR11 signaling (*Alexopoulou et al., 2001*; *Yarovinsky et al., 2005*; *Mathur et al., 2012*). However, cells expressing wildtype UNC93B1 responded robustly to the oligoribonucleotide sequence derived from *Staphylococcus aureus* (hereafter referred to as Sa ORN) that was recently shown to stimulate TLR13 (*Li and Chen, 2012*; *Oldenburg et al., 2012*). The response of UNC93B1-Y539A cells stimulated with Sa ORN was comparable to that of wildtype cells, suggesting that TLR13 does not require the UNC93B1/AP-2 pathway to access endosomal signaling compartments (*Figure 7A*).

We also examined trafficking of other UNC93B1-dependent TLRs biochemically in cells expressing UNC93B1-WT, UNC93B1-H412R, or UNC93B1-Y539A. TLR13 acquired EndoH-resistant glycans, and the glycosylation was absent in UNC93B1-H412R cells but unaffected in UNC93B1-Y539A expressing cells. Moreover, the full-length EndoH-resistant form did not accumulate in these cells, as we observed

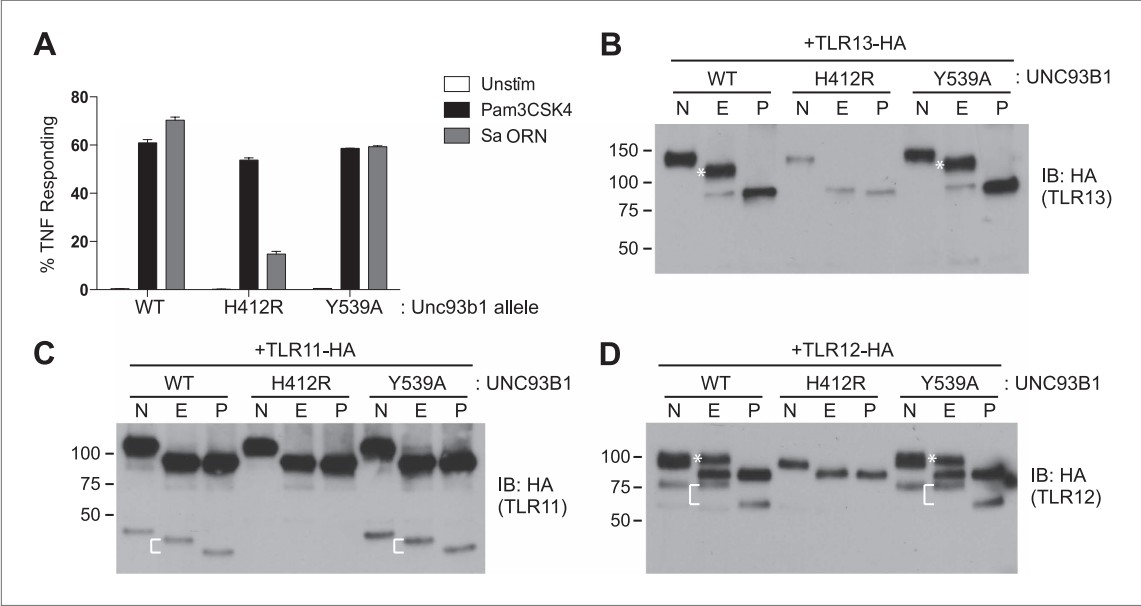

**Figure 7**. TLR11, TLR12, and TLR13 traffic independently of the UNC93B1/AP-2 pathway. (**A**) TLR13 signaling is unaffected by UNC93B1-Y539A. *3d* iMac cells expressing UNC93B1-WT, -H412R, or -Y539A were stimulated with 1 µg/ml Pam3CSK4 (black) or 1 nM Sa ORN complexed with DOTAP (grey) and harvested for intracellular TNFα staining 5 hr after stimulation. Percentages of TNF-producing cells are plotted. (**B**) TLR13 trafficking is unaffected by UNC93B1-Y539A. TLR13-HA was immunoprecipitated from *3d* iMac cells expressing UNC93B1-WT, -H412R, or -Y539A, treated with EndoH (E), PNGaseF (P) or left untreated (N), and visualized by anti-HA immunoblot. Asterisk indicates full length EndoH resistant form of the receptor. (**C**) and (**D**) TLR11 (**C**) and TLR12 (**D**) trafficking is unaffected by UNC93B1-Y539A. TLR11 and TLR12 was treated as in (**B**). Asterisk indicates full length EndoH resistant form of the receptor. The bracket indicates the migration difference between EndoH-treated and PNGaseF-treated cleaved form of the receptors. Results are representative of at least three experiments (**A**) or two experiments (**B**–**D**).

for TLR9. Together with the responses to TLR13 ligands, these results indicate that TLR13 traffics independently of the UNC93B1/AP-2 pathway. We performed similar experiments with TLR11 and TLR12, which both recognize profilin (*Yarovinsky et al., 2005*; *Koblansky et al., 2012*). Both of these receptors appear to undergo ectodomain cleavage, although the sizes of the cleaved receptors are quite distinct: ~45 kDa for TLR11 and ~75 kDa for TLR12 (*Figure 7C,D*). Importantly, both receptors acquired EndoH-resistant glycans, and the glycosylation was unaffected in UNC93B1-Y539A expressing cells. Thus, TLR11 and TLR12 also do not require the UNC93B1/AP-2 pathway for proper trafficking.

Altogether, these results suggest that TLR9 trafficking may be unusual among the endosomal TLRs by trafficking via the plasma membrane en route to endosomes. It is important to note, though, that we have not examined TLR3 trafficking. Thus, it remains possible that TLR3 may also utilize this route, especially considering the several reports that describe TLR3 surface expression (*Qi et al., 2010*, *2012*; *Pohar et al., 2012*; *Weber et al., 2012*).

## TLR7 and TLR9 association with UNC93B1 is mutually exclusive

We have shown that UNC93B1 associates with both the full-length and cleaved forms of TLR9 and TLR7 (*Figures 3E, 6C*); however, the differential post-Golgi trafficking routes taken by TLR7 and TLR9 suggest that UNC93B1 must associate with TLR7 and TLR9 in separate complexes. To test this possibility directly, we examined interactions between TLR7, TLR9, and UNC93B1 by immunoprecipitation. While both TLR7 and TLR9 could precipitate UNC93B1, we could never detect interactions between the two receptors (*Figure 8*). Thus, UNC93B1 binding to TLR7 appears to preclude binding to TLR9 and vice versa. These data support the idea that endosomal TLRs compete for binding to UNC93B1 in order to exit from the ER (*Wang et al., 2006*; *Fukui et al., 2009*, *2011*).

## Discussion

The studies presented here address several poorly understood aspects of TLR cell biology. First, we describe mechanisms by which UNC93B1 can regulate multiple aspects of TLR trafficking. UNC93B1

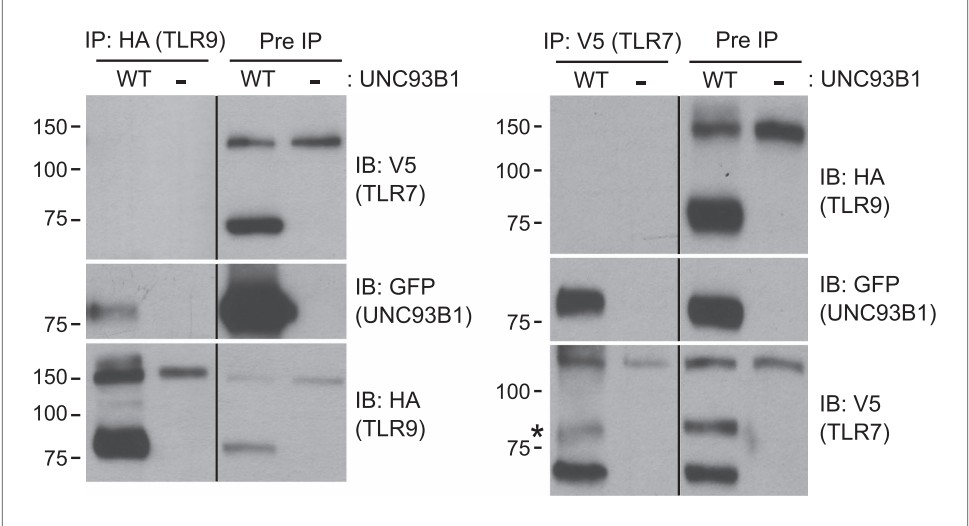

**Figure 8**. TLR7 and TLR9 association with UNC93B1 is mutually exclusive. TLR9 and TLR7 associate with UNC93B1 in distinct complexes. *3d* iMac cells expressing TLR9-HA, TLR7-V5-His, and WT UNC93B1-GFP (WT) or no UNC93B1 (−) were lysed in TNT buffer conditions then incubated with anti-HA matrix or anti-V5 protein A/G beads. Immunoprecipitated proteins were analyzed by SDS-PAGE and immunoblotted with HA, GFP and V5 antibodies. Asterisk indicates remaining UNC93B1-GFP signal. Results are representative of two experiments.

controls exit of at least six TLRs (TLR3, TLR7, TLR9, TLR11, TLR12 and TLR13) from the ER by regulating their loading into COPII vesicles. In addition, we show that UNC93B1 acts at the cell surface by recruiting clathrin AP-2 to internalize TLR9 from the plasma membrane into endosomes. Perhaps most surprisingly, our results indicate that TLR7 and TLR9 have different requirements for UNC93B1 and demonstrate that the localization of these receptors is controlled through distinct post-Golgi trafficking mechanisms. This last point may provide an explanation for the contrasting roles played by TLR7 and TLR9 in SLE.

Previous work has demonstrated that TLR9 and TLR7 fail to reach endosomes in the absence of functional UNC93B1 (*Kim et al., 2008*). Interactions between UNC93B1 and TLRs appear necessary for proper trafficking (*Brinkmann et al., 2007*), but until our work it was unclear how UNC93B1 mediated delivery of TLRs to endosomal compartments. The most direct examination of UNC93B1 function concluded that the protein translocates with TLRs from the ER to endolysosomes without passing through the Golgi apparatus (*Brinkmann et al., 2007*; *Kim et al., 2008*). In contrast, we propose that UNC93B1 is required for loading of TLR9 into COPII vesicles, which direct transport from the ER to the cis-Golgi (*Zanetti et al., 2012*). Multiple lines of evidence support the conclusion that UNC93B1 regulates this essential aspect of ER exit. First, UNC93B1 is detected within COPII vesicles, but the nonfunctional UNC93B1-H412R mutant is not. Second, a fraction of UNC93B1 protein acquires EndoH-resistant glycans, indicative of trafficking through the medial-Golgi. Again, the UNC93B1-H412R mutant does not acquire EndoH resistance. Third, in cells lacking UNC93B1, TLR9 is not detected within COPII vesicles. Fourth, the EndoH-resistant precursor and cleaved forms of TLR9 are absent in cells not expressing functional UNC93B1. Importantly, we ruled out the possibility that UNC93B1 functions as a folding chaperone by showing that CD4-TLR fusion proteins require UNC93B1 for ER exit. Altogether, these results argue that UNC93B1 regulates TLR passage through the general secretory pathway, instead of controlling direct translocation between the ER and endosomes. We cannot formally rule out that a pool of UNC93B1 uses this non-canonical route, but it would seem to have little if any relevance for TLR trafficking. The recent identification of Sec22b as a factor required for ER to endosome translocation may allow for further investigation of this possibility (*Cebrian et al., 2011*).

Our work suggests that at least six, and possibly seven, TLRs are regulated by UNC93B1 in mice. TLR3, TLR7, TLR9, TLR11, TLR12 and TLR13 require UNC93B1 for exit from the ER. It is likely that TLR8 also requires UNC93B1 (*Itoh et al., 2011*), based on its similarity to TLR7, but we have not yet tested this possibility. Regulation of ER exit of TLR11, TLR12 and TLR13 by UNC93B1 is not completely

unexpected, as these TLRs were previously linked to UNC93B1 (*Brinkmann et al., 2007*; *Melo et al., 2010*; *Pifer et al., 2011*; *Koblansky et al., 2012*; *Oldenburg et al., 2012*). The determinants of UNC93B1 binding to each of these TLRs remain poorly defined. Residues within the first 50 amino acids of UNC93B1 appear necessary for export of TLR9 but not for export of TLR7. This result suggests that distinct regions of UNC93B1 are required for association with TLR9 and TLR7, although measuring how these mutations impact TLR/UNC93B1 interactions has been challenging for us and for other groups (*Fukui et al., 2009*). The fact that UNC93B1-D34A-expressing mice exhibit enhanced TLR7 responses and develop SLE suggests that UNC93B1 is limiting in the ER, at least for TLR7 (*Fukui et al., 2011*). Presumably, the inability of UNC93B1-D34A to interact with TLR9 results in greater TLR7 export and enhanced signaling. These results and our finding that TLR7 and TLR9 association with UNC93B1 is mutually exclusive suggest that each UNC93B1 molecule may interact with a single TLR. This specificity is particularly important in light of our findings that trafficking routes of TLR7 and TLR9 are distinct. Whether UNC93B1 is similarly selective for other TLRs and identification of the domains that mediate any selectivity are important topics for future studies. Selective export of individual TLRs may provide a mechanism for differential regulation of endosomal TLR responses between cell types or in response to external signals.

Through analysis of UNC93B1 mutants we identified an additional role for UNC93B1 in directing post-Golgi trafficking of TLR9. Unlike typical COPII loading factors, UNC93B1 remains associated with its cargo, traffics with TLR9 to the cell surface, and associates with AP-2 via a YXXΦ motif (*Figure 9*). Recruitment of AP-2 is necessary for internalization of TLR9 and subsequent trafficking to endolysosomes. This AP-2-dependent trafficking route has been described for several proteins localized to endolysosomes, including LAMP-1, LAMP-2, and MHC Class II (*Gough et al., 1999*; *Janvier and Bonifacino, 2005*; *McCormick et al., 2005*). In some cases, the same protein can access endocytic compartments through multiple routes. For example, LAMP-1/2 can also traffic directly from the TGN to endosomes (*Karlsson and Carlsson, 1998*). While we cannot rule out that TLR9 may reach endolysosomes through multiple routes, AP-2-mediated internalization appears to be the main pathway of delivery, at least in the cells types we have examined. It is interesting that TLR9 and UNC93B1 would have evolved such dependency on this route, especially considering the potential for self-DNA recognition associated with surface localization of TLR9 (*Barton et al., 2006*; *Mouchess et al., 2011*). Indeed, our results appear to underscore why the requirement for ectodomain processing is a critical mechanism ensuring that receptors at the plasma membrane remain nonfunctional (*Ewald et al., 2008*; *Mouchess et al., 2011*).

One of the most exciting aspects of our study is the observation that trafficking of TLR7 and TLR9 are distinct. We find no evidence that TLR7 requires UNC93B1 for recruitment of AP-2. Instead, TLR7 appears to employ AP-4 in a direct route of traffic from the TGN to the endosome. These results provide the first evidence that different pathways control TLR7 and TLR9 trafficking and localization. One potential implication of this result is that these pathways may deliver TLR9 and TLR7 to distinct compartments with different access to ligands or distinct signaling properties. We find that TLR11, TLR12 and TLR13, like TLR7, do not require the UNC93B1/AP-2 pathway. However, it remains to be determined whether these TLRs utilize the AP-4 pathway or another pathway to traffic from TGN to endosomes. Further compartmental specialization is generated by AP-3, which interacts with TLR9 and directs the receptor to endosomal compartments dedicated to type I interferon signaling (*Honda et al., 2005*; *Blasius et al., 2010*; *Sasai et al., 2010*). TLR7 may also utilize AP-3 to reach this specialized compartment, although we could not detect interaction between AP-3 and TLR7 (*Figure 6E*). Whether there is transport of TLRs between each of the compartments serviced by AP-2, AP-3, and AP-4 remains an open question.

The use of distinct molecular pathways to regulate endosomal TLRs may allow for differential regulation of trafficking, either in response to external cues or between different cell types. In addition, TLR7- and TLR9-containing compartments may have differing abilities to access internalized ligands, influencing responses to microbial or self ligands. These possibilities are particularly intriguing when considering the contrasting roles played by TLR7 and TLR9 in SLE, where loss of TLR7 protects against disease while loss of TLR9 exacerbates disease (*Christensen et al., 2006*). Our findings raise the possibility that distinct cell biological regulation may underlie the different roles played by these receptors in autoimmune disease. Defining the mechanisms underlying this regulation may help explain the etiology of certain autoimmune diseases as well as provide opportunities to selectively manipulate distinct aspects of TLR activation.

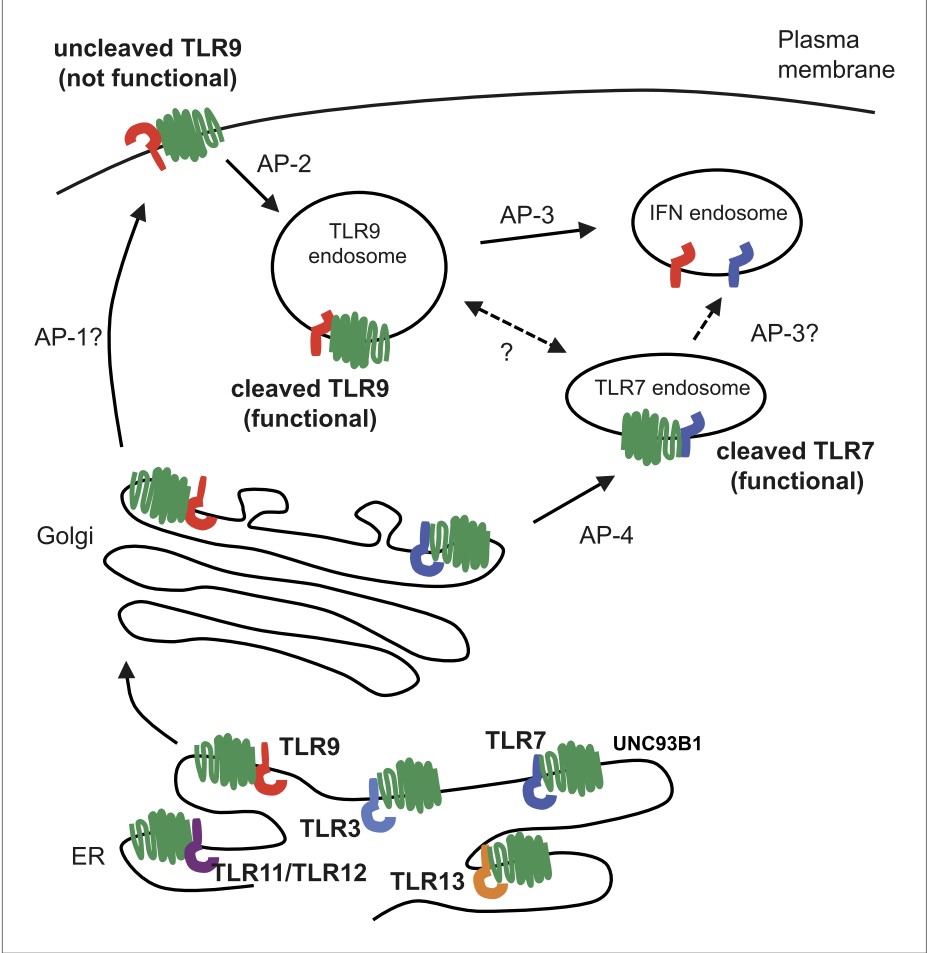

**Figure 9**. Trafficking pathways controlling localization of endosomal TLRs. UNC93B1 interacts with several TLRs in the ER and facilitates loading into COPII vesicles. Unlike typical COPII loading factors, UNC93B1 remains associated with TLR9 and TLR7 after exit from the ER. Through its recruitment of AP-2, UNC93B1 is necessary for endocytosis of TLR9 from the plasma membrane into endosomes. TLR7 does not rely on this trafficking route. Instead, TLR7 utilizes AP-4 to bypass the cell surface and traffic directly to endosomes. This difference in trafficking may result in TLR9 and TLR7 accessing distinct compartments with unique functional properties related to the function of each receptor. DOI: 10.7554/eLife.00291.011

## Materials and methods

### Antibodies and reagents

The following antibodies were used for immunoblots, immunoprecipitations, or flow cytometry: anti-HA as purified antibody or matrix (3F10; Roche, Indianapolis, IN), anti-FLAG as purified antibody or matrix (M2 and M5; Sigma-Aldrich, St. Louis, MO), anti-GFP (JL-8; Clontech Laboratories, Inc, Mountain View, CA), anti-GFP as purified or matrix (RQ2; MBL International Corporation, Woburn, MA), anti-*myc* (purified mouse; Invitrogen, Grand Island, NY), anti-V5 (mouse monoclonal; Invitrogen) anti-CD4 (RM4-5; BD Biosciences, San Jose, CA), anti-Lamp-1 (1D4B; BD Biosciences), anti-calnexin (rabbit polyclonal; Enzo Life Sciences, Farmingdale, NY), anti-ERGIC/p58 (rabbit) has been previously described (*Merte et al., 2010*), anti-TNFα-PE or -APC (MP6-XT22; eBiosciences, San Diego, CA), anti-CD71-APC (OKT9; eBiosciences), goat anti-mouse IgG-AlexaFluor647 (Invitrogen), goat anti-rat-HRP, sheep anti-mouse-HRP, and donkey anti-rabbit-HRP (GE Healthcare, Waukesha, WI). For immunofluorescence: rabbit anti-HA (Y11; Santa Cruz Biotechnology, Dallas, TX), goat anti-mouse Cy3 (Jackson Immunoresearch, West Grove, PA), goat anti-rabbit-AlexaFluor647 (Invitrogen). The following TLR ligands were used to stimulate cells: CpG ODN (TCCATGACGTTCCTGACGTT, all phosphorothioate linkages) and Sa ORN ('Sa17'; *Oldenburg et al., 2012*) (GACGGAAAGACCCCGUG

RNA sequence purchased from Integrated DNA Technologies, San Diego, CA), R848 (InvivoGen, San Diego, CA), and Pam3CSK4 (InvivoGen). 3× FLAG peptide was purchased from Sigma-Aldrich. Digitonin was purchased from Wako Pure Chemical Industries, Ltd. (Richmond, VA) or Calbiochem (Billerica, MA). Lipofectamine-LTX reagent (Invitrogen) was used for transient transfection of plasmid DNA. Lipofectamine RNAiMAX reagent (Invitrogen) was used for siRNA delivery. DOTAP liposomal transfection reagent (Roche) was used for transfection of Sa ORN in PBS. OptiMEM-I (Invitrogen) was used as media to form nucleic acid complexes for transient transfections.

## Mice

*Unc93b1*[3d/3d] mice (*Tabeta et al., 2006*) were obtained from the MMRRC at University of California, Davis. C57Bl/6 were purchased from The Jackson Laboratory (Bar Harbor, ME). All mice were housed in the animal facilities at the University of California, Berkeley according to guidelines of the Institutional Animal Care and Use Committee.

## Plasmid constructs

Pfu Turbo polymerase (Agilent Technologies, Santa Clara, CA) was used according to manufacturer's instructions for site directed mutagenesis. The following mouse stem cell virus (MSCV)-based retroviral vectors were used to express UNC93B1, TLR9, and TLR7 in cell lines: MSCV2.2 (IRES-GFP), MSCV-Thy1.1 (IRES Thy1.1), MIGR2 (IRES-hCD2). The following epitope tags were fused to the C-terminus of UNC93B1: 3× FLAG (DYKDHDGDYKDHDIDYKDDDDK), Myc (EQKLISEEDL), HA (YPYDVPDYA) and entire eGFP cDNA derived from pIRES-eGFP plasmid (Clontech). TLR9 was fused to HA at the C-terminal end or with 3× FLAG at the N-terminal end as previously described (*Ewald et al., 2008*; *Mouchess et al., 2011*). TLR7 sequence was synthesized after codon optimization by Invitrogen's GeneArt Gene Synthesis service and cloned into same vectors as TLR9 and tagged with C-terminal HA or C-terminal V5-His (GKPIPNPLLGLDST-HHHHHH). TLR11, TLR12, TLR13 was C-terminally tagged with HA and cloned into MSCV2.2. UNC93B1 shRNA and control were generated in MSCV-Lmp, as previously described (*Ewald et al., 2008*). CD4-TLR chimeras were generated in pCDNA3.1 (Invitrogen). CD4 extracellular domain (mouse 1–390 a.a.) was fused to transmembrane domain and cytosolic regions of the following TLRs and C-terminally tagged with HA: TLR4 (620–835 a.a.), TLR9 (mouse 803–1032 a.a.), TLR3 (human 691–904 a.a.), TLR7 (human 825–1049), TLR11 (mouse 703–926 a.a.), TLR13 (mouse 770–991 a.a.). Rat AP-2μ-HA containing an internal HA tag in pCDNA3 was provided by A. Sorkin (while at the University of California, San Diego, CA, now at the University of Pittsburgh, PA) (*Nesterov et al., 1999*).

For yeast-two-hybrid assays, N-terminal (1–59 a.a.) and C-terminal (515–598 a.a.) cytosolic regions of UNC93B1 were fused to Gal4 DNA binding domain (DBD) by cloning into pGBT9 (Clontech). TLR7 cytosolic region (mouse 862–1061 a.a.), TLR9 cytosolic region (mouse 838–1032 a.a.), fused to Gal4-DBD by cloning into pGBT9. AP-1Aμ, AP-2μ, AP-3Aμ, AP-3Bμ, AP-4μ were cloned into pACT2 (Clontech) were provided by J. Bonifacino (National Institutes of Health, Bethesda, MD).

## Cell lines and tissue culture conditions

HEK293T cells were obtained from American Type Culture Collection (ATCC, Manassas, VA). GP2-293 packaging cell lines were obtained from Clontech. Phoenix-Eco (ØNX-E) cells were provided by G. Nolan (Stanford University, Palo Alto, CA). Mouse embryonic fibroblasts (MEFs) are TLR2/TLR4 double knockout genotype immortalized with SV40 large T-antigen. COS7 were obtained from the Berkeley cell culture facility. The above cell lines were cultured in DMEM supplemented with 10% (vol/vol) FCS, L-glutamine, penicillin-streptomycin, sodium pyruvate, and HEPES (pH 7.2) (Invitrogen). RAW264 macrophage cell lines (ATCC) and immortalized macrophages (generated as described below) were cultured in RPMI 1640 (same supplements as above).

To generate immortalized macrophage cell lines, bone marrow from *Unc93b1*[3d/3d] mice was cultured in RPMI 1640 media supplemented with supernatant containing M-CSF, as previously described (*Arpaia et al., 2011*), as well as virus encoding both v-raf and v-myc (*Blasi et al., 1985*). After 8 days, macrophages were removed from M-CSF-containing media and cultured in RPMI 1640 media with added supplements as described above.

## Retroviral transduction

For retroviral transduction of immortalized macrophages, VSV-G-psuedotyped retrovirus was made in GP2-293 packaging cells (Clontech). GP2-293 cells were transfected with retroviral vectors and

pVSV-G using Lipofectamine LTX reagent. 24 hr post-transfection, cells were incubated at 32°C. 48 hr post-transfection viral supernatant (with polybrene at final 5 µg/ml) and was used to infect target cells overnight at 32°C and protein expression was checked 48 hr later. Target cells were sorted on MoFlo Beckman Coulter Sorter to match expression.

For retroviral transduction of bone marrow derived macrophages, retrovirus was produced with the ØNX-E packaging line. Bone marrow cells were transduced with viral supernatant on two successive days while cultured in M-CSF containing RPMI media until harvested on day 8.

## Luciferase assays

Activation of NF-κB in HEK293T cells was performed as previously described (*Ewald et al., 2008*). Briefly, transfections were performed in OptiMEM-I (Invitrogen) with LTX transfection reagent (Invitrogen) according to manufacturer's guidelines. Cells were stimulated with 1–10 µg/ml R848 after 24 hr and lysed by passive lysis after an additional 12–16 hr. Luciferase activity was measured on a LMaxII-384 luminometer (Molecular Devices, Sunnyvale, CA).

## Lysate preparation, SDS-PAGE, immunoblotting

Cell lysates were prepared with TNT buffer (20 mM Tris [pH 8.0], 200 mM NaCl, 1% Triton X-100, 4 mM EDTA and supplemented with EDTA-free complete protease inhibitor cocktail; Roche) unless otherwise noted. Digitonin lysis buffer (50 mM Tris [pH 7.4], 150 mM NaCl, 5 mM EDTA [pH 8.0], 1% Digitonin added fresh and supplemented with EDTA-free complete protease inhibitor cocktail) was used for co-immunoprecipitations. Lysates were cleared of insoluble material by centrifugation. For immunoprecipitations, lysates were incubated with anti-HA matrix, anti-FLAG matrix, anti-GFP matrix or with purified antibody conjugated to Protein A/G beads (ThermoFisher Pierce, Rockford, IL) and precipitated proteins were denatured in SDS-PAGE buffer separated by SDS-PAGE (Tris–HCl self cast gels or Bio-Rad TGX precast gels [Bio-Rad, Hercules, CA]), and probed by the indicated antibodies. For anti-FLAG matrix immunoprecipitations, 3× FLAG peptide (Sigma-Aldrich) was used to elute.

## Endoglycosidase H assay

Immunoprecipitated proteins or total lysate were denatured and treated with Endoglycosidase H or PNGase F according to manufacturer's instructions. All enzymes and buffers were purchased from New England Biolabs (Ipswich, MA).

## Flow cytometry

To measure TNFα production, we added brefeldinA to cells 30 min after stimulation, and cells were collected after an additional 4 hr, and cells were stained for intracellular cytokines with a Fixation & Permeabilization kit according to manufacturer's instructions (eBioscience).

For FLAG-TLR surface expression, HEK293T cells stably expressing N-terminally tagged 3× FLAG-TLR9 were stained with anti-FLAG (M5; Sigma-Aldrich) antibody followed by Alexa 647 goat anti-mouse IgG secondary antibody (Invitrogen). All data were collected on LSR II (Becton Dickinson, Franklin Lakes, NJ) or FC-500 (Beckman Coulter, Indianapolis, IN) flow cytometers and were analyzed with FloJo software (TreeStar, Inc. Ashland, OR).

## Microscopy

Co-localization studies were performed on Leica TCS confocal microscope. The images were taken with a 40× oil immersion objective and treated with 2× digital zoom. All images were processed by Adobe Photoshop. Cells were allowed to settle overnight on coverslips. Coverslips were washed with PBS, fixed with 4% paraformaldehyde/PBS, and permeabilized with 0.5% saponin/PBS. Slides were incubated in freshly made 0.1% sodium borohydride 0.1% saponin before being stained in 1% Bovine Serum Albumin (Fisher Scientific, Pittsburgh, PA)/0.1% saponin in PBS with rabbit anti-HA (Santa Cruz), rat anti-Lamp-1 (BD Biosciences) then with secondary antibodies, goat anti-Rabbit 647 (Invitrogen) and goat anti-Rat Cy3 (Jackson Immunoresearch).

## Phagosome isolation

RAW264 cells were used to isolate phagosomes as previously described (*Ewald et al., 2008*). Briefly, cells were incubated with 2 µM latex beads (Polysciences, Inc., Warrington, PA) for 1 hr. Cells were disrupted by dounce homogenization to release intact phagosomes. Following centrifugation in sucrose step gradient, phagosomes were harvested from the 20–10% sucrose interface. Lysates were analyzed by immunoblot.

## siRNA knockdown

$3.5 \times 10^5$ HEK293Ts were plated in 2 ml antibiotic free media per well in six-well plates and reverse transfected with 5 µl 20 µM siRNA in 500 µl OptiMEM-I and 5 µl of Lipofectamine RNAiMAX for 48–96 hr until harvest for flow cytometry or SDS-PAGE and immunoblot analysis. siRNA duplexes against human AP-2µ were purchased from ThermoFisher Dharmacon RNAi Technologies (Waltham, MA) with the following sequence: 5'-GGAGAACAGUUCUUGCGGC-3' and with the following conditions: ON-TARGET - Enhanced Antisense Loading, Standard (A4), UU added to 3' end. Control siRNA (ON-TARGETplus Non-targeting siRNA #1) was purchased from Dharmacon.

## Yeast two-hybrid interaction

Cells of an overnight culture (2.5 ml) of yeast strain PJ69-4a in YPD media grown at 30°C were washed in $H_2O$ and mixed sequentially with the following: 50% PEG-3500 (Sigma-Aldrich), 10 mM lithium acetate in Tris-EDTA (TE) (pH 7.5), denatured salmon sperm DNA (Invitrogen) and 100 ng of plasmid constructs, then mixed. Yeast were incubated for 30 min at 30°C and heat-shocked for 15 min at 42°C. Cells were centrifuged, resuspended in $H_2O$, and spread on YNB plates with –Trp –Leu dropout mix. Liquid cultures in Trp-Leu broth were grown at 30°C overnight. Cells were normalized to 1.0 OD and 1:10 dilutions made. 4 µl of each dilution was plated on –Trp–Leu–His, and –Trp–Leu–Ade at 30°C. YPD, YNB and dropout mixtures purchased from Sunrise Science. Growth on –Trp–Leu–His or –Trp–Leu–Ade plates was recorded at day 3.

## In vitro COPII budding assay

COPII vesicle formation was performed as described previously (*Kim et al., 2005*; *Merte et al., 2010*). In brief, RAW264 cells grown in 10 × 10-mm plates or COS7 cells grown in 6 × 100-mm plates were washed in PBS, removed from plates with trypsin, and washed again in PBS containing 10 µg/ml soybean trypsin inhibitor. Cells were permeablized with 40 µg/ml digitonin for 5 min in ice-cold KHM buffer (110 mM KOAc, 20 mM Hepes pH 7.2 and 2 mM Mg(OAC)$_2$) and washed and resuspended in 100 µl KHM. Each reaction contained KHM and where indicated an ATP regenerating system (40 mm creatine phosphate, 0.2 mg/ml creatine phosphokinase, and 1 mm ATP), 0.2 mm GTP, and rat liver cytosol (prepared as described previously; *Kim et al., 2005*). Reactions were incubated at 30°C for 60 min. A 75-µl aliquot of the vesicle fraction was separated from the donor microsomal fraction by centrifugation at 14,000×$g$ for 20 min at 4°C. Donor fraction was lysed in 75 µl of Buffer C (10 mm Tris–HCl [pH 7.6], 100 mm NaCl, 10% [w/v] SDS plus protease inhibitor mixture). The vesicles were collected by centrifugation at 50,000 rpm at 4°C in a Beckman TLA100 rotor for 30 min. Isolated vesicles were lysed in 20 µl of Buffer C. Donor membrane (20% total) and isolated vesicles (75% of total) were separated by SDS-PAGE and analyzed by immunoblotting.

## Acknowledgements

We thank members of the Barton, Schekman, and Vance labs for helpful discussions and advice. We thank Hector Nolla for assistance with cell sorting at the Flow Cytometry Facility of the Cancer Research Laboratory at UC Berkeley. RS is supported as an Investigator of the Howard Hughes Medical Institute and as a Senior Fellow of the UC Berkeley Miller Institute.

# Additional information

### Competing interests

RS: Editor-in-Chief, *eLife*. The other authors declare that no competing interests exist.

### Funding

| Funder | Grant reference number | Author |
| --- | --- | --- |
| National Institutes of Health | AI072429, AI063302 | Gregory M Barton |
| Lupus Research Institute | | Gregory M Barton |
| Burroughs Wellcome Fund | | Gregory M Barton |
| Weisenfeld Fellowship | | Bettina L Lee |

The funders had no role in study design, data collection and interpretation, or the decision to submit the work for publication.

## Author contributions

BLL, Conception and design, Acquisition of data, Analysis and interpretation of data, Drafting or revising the article; JEM, JHS, LY, ZRN, Acquisition of data; RS, Drafting or revising the article; GMB, Conception and design, Analysis and interpretation of data, Drafting or revising the article

## Ethics

Animal experimentation: All animal experiments were performed in accordance with the Guide for the Care and Use of Laboratory Animals of the National Institutes of Health. Experimental protocols were reviewed and approved by the University of California Animal Care and Use Committee (protocol number R298).

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
