## [Decision Letter]

Thank you for choosing to send your work entitled “Unc93b1 mediates differential trafficking of TLR7 and TLR9” for consideration at *eLife*. Your article has been evaluated by a Senior editor and 3 reviewers, one of whom is a member of our Board of Reviewing Editors.

The Reviewing editor and the other reviewers discussed their comments before we reached this decision, and the Reviewing editor has assembled the following comments based on the reviewers' reports.

This is an excellent study providing novel insights into the mechanisms of differential trafficking of TLR7 and TLR9 by the multipass transmembrane protein Unc93b1. The study will have general impact in immunology and cell biology. On the basis of enthusiastic comments by the Reviewing editor and two other reviewers, we will be happy to accept this paper for publication in *eLife*, provided that the authors address the minor issues raised by the reviewers:

1) A striking finding of this study is the differential use of the Unc93B/AP-2 trafficking pathway by TLR9 and TLR7. The authors suggest that TLR7 can direct itself to endosomes from the golgi through interactions with AP-4 subunits. Is this a common feature of the several TLRs that bind to Unc93B? If yeast 2-hybrid analyses are performed with other endosomal TLRs, are there other examples of TLR-AP4 interactions, or is TLR7 unique?

2) The authors make the intriguing suggestion that Unc93B binds to a single TLR at a time. This model predicts that at least five different complexes of Unc93B exist in cells (Unc-TLR9 complexes, Unc-TLR7 complexes, etc). If this is true, then immunoprecipitation assays with TLR7 should not be able to co-purify TLR9, while Unc93B pulldowns should isolate both TLR7 and TLR9. The authors are encouraged to perform these assays.

3) In Figure 7, the schematic figure indicates the role of AP-1 in transfer of TLR9 from Golgi to the plasma membrane. However, we could not find data for this in the study. Can the authors clarify this?

---

## [Author Response]

*1) A striking finding of this study is the differential use of the Unc93B/AP-2 trafficking pathway by TLR9 and TLR7. The authors suggest that TLR7 can direct itself to endosomes from the golgi through interactions with AP-4 subunits. Is this a common feature of the several TLRs that bind to Unc93B? If yeast 2-hybrid analyses are performed with other endosomal TLRs, are there other examples of TLR-AP4 interactions, or is TLR7 unique*?

To address whether other UNC93B1-dependent TLRs require AP-2 versus AP-4 to reach endosomes, we examined whether signaling is affected in macrophages expressing the UNC93B1-Y539A mutant, which fails to recruit AP-2. As shown in our new Figure 7A, the response to TLR13 ligands is unaffected in these cells, suggesting that TLR13 does not require the UNC93B1/AP-2 pathway for its function. Additionally, we present biochemical evidence that TLR13 trafficking in UNC93B1-Y539A mutant cells is unimpaired (Figure 7B). This result suggests that TLR13 (like TLR7) may utilize AP-4 for Golgi to endosome trafficking, although we do not yet have any direct evidence for this conclusion.

We also tested how responses to TLR3 (PolyI:C) and TLR11 (profilin and flagellin) ligands were affected by the UNC93B1-Y539A mutant. Unfortunately, we were unable to detect responses to TLR3 and TLR11 ligands in macrophages, even when we ectopically expressed TLR11. However, we were able to examine TLR11 and TLR12 trafficking in UNC93B1-Y539A mutant cells and found that the trafficking of these receptors is also unaffected by disruption of the UNC93B1/AP-2 pathway (Figure 7C, D).

As suggested by the reviewers, we have also tried to detect interactions between the cytosolic domains of other endosomal TLRs and the μ subunit of AP-4 using the yeast 2-hybrid (Y2H) assay. We failed to detect any interactions between AP-4 and TLR3, TLR9, TLR11, or TLR13. While this result could support the conclusion that trafficking of TLR3, TLR11, and TLR13 is distinct from trafficking of TLR7, we are hesitant to leap to this conclusion. Our primary concern is that, for reasons that remain unclear, TLR cytosolic domains do not behave equivalently in Y2H assays. For example, Brown et al have reported that the TLR9 cytosolic domain fails to interact with MyD88 by Y2H, while TLR2, TLR5, TLR7, and TLR8 do interact with MyD88 (see Brown et al, EJI, 2006). We have experienced the same problems with the TLR9 cytosolic domain, which is why we did not include Y2H data for TLR9, and suspect that TLR3, TLR11, and TLR13 have similar limitations. Therefore, we cannot rule out that lack of interaction between TLR3, TLR11, or TLR13 (or TLR9 for that matter) and AP-4 is due to a technical limitation of the Y2H assay for these TLRs. For this reason, we have relied on the functional assays that we discussed previously. We would prefer to omit the AP-4 Y2H interaction data from our manuscript due to these technical issues, but we have included the results below for the benefit of the reviewers. [Editors' note: the authors have agreed to make the results available in the author response, with the caveats described above.]Author response image 1.Results from a yeast two-hybrid assay testing for interaction between the AP-4μ subunit and the C-terminal cytosolic region of TLR3, 7, 9, 11, 13.Growth on –His–Trp– Leu plates (–His) indicates interaction. Growth on –Trp–Leu plates (+His) serves as a control.

*2) The authors make the intriguing suggestion that Unc93B binds to a single TLR at a time. This model predicts that at least five different complexes of Unc93B exist in cells (Unc-TLR9 complexes, Unc-TLR7 complexes, etc). If this is true, then immunoprecipitation assays with TLR7 should not be able to co-purify TLR9, while Unc93B pulldowns should isolate both TLR7 and TLR9. The authors are encouraged to perform these assays*.

We have performed these biochemical experiments and include these data in our new Figure 8. As predicted by the model, immunoprecipitation of TLR7 or TLR9 does precipitate UNC93B1, but does not precipitate the other receptor.

*3) In Figure 7, the schematic figure indicates the role of AP-1 in transfer of TLR9 from Golgi to the plasma membrane. However, we could not find data for this in the study. Can the authors clarify this*?

Including AP-1 in the schematic was a mistake. While we think it is possible that AP-1 mediates this step, we do not have evidence for or against this possibility. We have corrected the schematic (Figure 9) to more accurately reflect our current knowledge.

Unrelated to the three points raised by the reviewers, we have also included additional evidence that AP-4 is involved in TLR7 trafficking to endosomes. In our original submission we described a mutation in TLR7, Y892A that disrupts AP-4 interaction. This mutant receptor no longer responds to TLR7 ligands. In this revised manuscript we now show that this receptor has a substantial reduction in ectodomain processing, further supporting the conclusion that failure to recruit AP-4 leads to reduced TLR7 trafficking to endosomes (Figure 6H).